# IMPLICIT JACOBIAN REGULARIZATION WEIGHTED WITH IMPURITY OF PROBABILITY OUTPUT

## ABSTRACT

Gradient descent (GD) plays a crucial role in the success of deep learning, but it is still not fully understood how GD finds minima that generalize well. In many studies, GD has been understood as a gradient flow in the limit of vanishing learning rate. However, this approach has a fundamental limitation in explaining the oscillatory behavior with iterative catapult in a practical finite learning rate regime. To address this limitation, we rather start with strong empirical evidence of the plateau of the sharpness (the top eigenvalue of the Hessian) of the loss function landscape. With this observation, we investigate the Hessian through simple and much lower-dimensional matrices. In particular, to analyze the sharpness, we instead explore the eigenvalue problem for the low-dimensional matrix which is a rank-one modification of a diagonal matrix. The eigendecomposition provides a simple relation between the eigenvalues of the low-dimensional matrix and the impurity of the probability output. We exploit this connection to derive sharpness-impurity-Jacobian relation and to explain how the sharpness influences the learning dynamics and the generalization performance. In particular, we show that GD has implicit regularization effects on the Jacobian norm weighted with the impurity of the probability output.

## 1 INTRODUCTION

Deep learning has shown to be powerful for many learning tasks in various areas. There has been a lot of work to understand how the learning algorithm leads to this successful training of deep neural networks. Especially, it is crucial to understand the geometric properties of the loss landscape of neural networks and their interaction with the gradient-based optimization methods, such as Stochastic Gradient Descent (SGD), along the training trajectory. It has been studied both from the optimization (Gur-Ari et al., 2018; Jastrzębski et al., 2019; Ghorbani et al., 2019; Liu et al., 2020; Lewkowycz et al., 2020; Cohen et al., 2021) and generalization (Hochreiter & Schmidhuber, 1997; Keskar et al., 2017; Dinh et al., 2017; Jastrzębski et al., 2017; Wang et al., 2018; Chaudhari et al., 2019; Fort et al., 2019; Jiang et al., 2020; Barrett & Dherin, 2021; Smith et al., 2021) point of view.

We aim at investigating the Hessian of the training loss (with respect to model parameter) and its top eigenvalue (also called sharpness). The sharpness characterizes the dynamics of neural network training along the optimization trajectory and is correlated with the generalization capability. For example, the sharpness increases in the beginning, and after reaching a certain value, training dynamics becomes unstable, oscillating along the top eigenvector (Jastrzębski et al., 2019; Cohen et al., 2021). Moreover, the rapid increase in the sharpness of the loss landscape in the early phase significantly impacts the final generalization performance (Achille et al., 2019; Jastrzebski et al., 2020; Lewkowycz et al., 2020; Jastrzebski et al., 2021). However, the Hessian of a deep neural network is very high-dimensional which makes it difficult to analyze its eigensystem. Recently, some researchers studied the Hessian by exploiting tools in randomized numerical linear algebra (Sagun et al., 2017; Papyan, 2018; 2019; Ghorbani et al., 2019; Yao et al., 2020) and decomposition of the Hessian (Papyan, 2018; 2019; Fort & Ganguli, 2019).

In this paper, we present a new decomposition of the Hessian using eigendecomposition of low-dimensional matrices. From the eigensystem of the low-dimensional matrix, we can provide a simple and intuitive explanation on the relation between its eigenvalue and the probability output. This

enables us to explain how the sharpness of the loss landscape influences the learning dynamics and the generalization performance.

We summarize the main contributions of the paper as follows:

- We decompose the Hessian with low dimensional matrices, the *logit Hessian* and the logit-weight Jacobian (defined in Definition 1), and investigate the Hessian by the eigendecomposition of the logit Hessian which is a rank-one modification of a diagonal matrix.
- We provide connections between the top eigenvalue of the logit Hessian and the impurity of the probability output.
- We derive a relation between the sharpness, the top eigenvalue of the logit Hessian and the Jacobian. We call it *sharpness-impurity-Jacobian relation*.
- We explain how the sharpness of the loss landscape influences the learning dynamics and the generalization performance. In particular, we find that gradient-based optimizations have implicit effects on penalizing the Jacobian norm (*Implicit Jacobian Regularization*) in a certain phase of training (*Active Regularization Period*).

## 2 RELATED WORK

We summarize some works on the Hessian, learning dynamics, and generalization of neural networks. In particular, we point out the issue of approximating SGD by a stochastic differential equation (SDE) because a continuous flow cannot capture the oscillatory behavior of discrete updates with iterative catapult, which plays a key role in limiting the sharpness of the loss landscape.

**Decomposition of the Hessian**   Sagun et al. (2016; 2017) empirically found that the eigenvalue spectrum of the Hessian during training is composed of two parts, the bulk which is concentrated around zero and the outliers which are scattered positively away from zero. They showed the bulk depends on the size of the network, and the outliers depend on the data. In particular, the number of outliers matches the number of classes of the data. Further, Papyan (2019) proposed a three-level hierarchical decomposition of the Hessian matrix according to each class, logit coordinate, and example. However, with different decomposition, we analyze the Hessian from another point of view.

**SGD as a SDE**   In many studies, SGD has been understood as a SDE in the limit of vanishing learning rate (Mandt et al., 2017; Li et al., 2017b;a; Smith & Le, 2018; Chaudhari & Soatto, 2018; Jastrzębski et al., 2017; Zhu et al., 2019; Park et al., 2019). However, some theoretical concerns have been raised for such approximations (Yaida, 2019). Moreover, Barrett & Dherin (2021) argued that the SDE analysis in the limit of vanishing learning rate cannot explain the generalization benefits of finite learning rates and they proposed a modified gradient flow for finite learning rates. However, they still consider a continuous gradient flow and thus it has a fundamental limitation in explaining the oscillatory behavior with iterative catapult in a practical learning rate regime (Smith et al., 2021), which will be detailed in the following paragraph.

**Oscillatory catapult and the plateau of the sharpness**   Xing et al. (2018) investigated the roles of learning rate and batch size in SGD dynamics through interpolating the loss landscape between consecutive model parameters during training. They observed SGD explores the parameter space, bouncing between walls of valley-like regions. The large learning rate maintains a high valley height, and a small batch size induces gradient stochasity. They both help exploration through the parameter space with different roles in the training dynamics. Jastrzębski et al. (2019) empirically investigated the evolution of the sharpness (the top eigenvalue of the Hessian) along the whole training trajectory of SGD. They observed initial growth of the sharpness as loss decreases, reaching a maximum sharpness determined by learning rate and batch size, and then it decreases towards the end of training. Due to the initial increase of the sharpness, the SGD step becomes too large compared to the shape of the loss landscape. This is consistent with the valley-like structure shown in Xing et al. (2018). Lewkowycz et al. (2020) investigated simple theoretical models with a solvable training dynamics. They showed that, in their setup with a large learning rate, the loss initially increases while the sharpness decreases, then it converges to a flat minimum. This mechanism is called the catapult mechanism. Recently, Cohen et al. (2021) found that full-batch GD typically operates in a regime

called the Edge of Stability where the sharpness can no longer increase and stays near a certain value, and the training loss behaves nonmonotonically but decreases globally. This behavior of the optimization at the Edge of Stability can be seen as repeated catapult mechanisms. They explicitly marked the limit of the sharpness with $2/\eta$ ($\eta$ =learning rate).

To describe the aforementioned evolution of the sharpness, Fort & Ganguli (2019) developed a theoretical model based on a random matrix modelling. To build a simple random model, they introduced assumptions about gradients and Hessians that they are i.i.d isotropic Gaussian with zero mean with varying variance during training. While they focus on building a random model based on the observation, we rather aim to explain the underlying mechanisms.

**Implicit bias in SGD**    There have been many studies on the implicit bias in SGD (Neyshabur, 2017; Zhang et al., 2021; Soudry et al., 2018). We review the most relevant and recent ones. Jastrzebski et al. (2021) empirically showed that SGD implicitly penalizes the trace of the Fisher Information Matrix (FIM). They also showed the trace of FIM explodes in the early phase of training when using a small learning rate and called it catastrophic Fisher explosion. Barrett & Dherin (2021); Smith et al. (2021) demonstrated that SGD implicitly penalizes the norm of the total gradient and the non-uniformity of the minibatch gradients. We demonstrate that the (logit-weight) Jacobian plays an important role in the generalization performance in each case.

## 3    BACKGROUND

In this section, we provide some notations, basic equations and definitions for the following sections. Throughout the paper, we use the denominator layout notation for the vector derivatives, i.e., $\nabla_{\boldsymbol{v}}\boldsymbol{u} = \left(\frac{\partial \boldsymbol{u}_j}{\partial \boldsymbol{v}_i}\right)_{ij} \in \mathbb{R}^{v \times u}$ where $\boldsymbol{u} : \mathbb{R}^v \to \mathbb{R}^u$ and $\boldsymbol{v} \in \mathbb{R}^v$. It is also generalized to the cases of scalar, $u = 1$ or $v = 1$.

We consider a problem of learning a $C$-class classifier which maps an input $\boldsymbol{x} \in \mathcal{X} \subset \mathbb{R}^d$ to a target label $y \in [C]$ where $[C] = \{1, 2, \cdots, C\}$. To this end, we build a parameterized model $f_{\boldsymbol{\theta}} : \mathcal{X} \to \mathcal{Z} \subset \mathbb{R}^C$ with a model parameter $\boldsymbol{\theta} \in \Theta \subset \mathbb{R}^m$ which outputs a logit vector $\boldsymbol{z} \equiv f_{\boldsymbol{\theta}}(\boldsymbol{x}) \in \mathcal{Z} \subset \mathbb{R}^C$ (we often omit the dependence on the input $\boldsymbol{x}$ and the model parameter $\boldsymbol{\theta}$). Then, the logit vector $\boldsymbol{z}$ is given as input to the softmax function to yield a probability vector $\boldsymbol{p} = \text{softmax}(\boldsymbol{z}) \in \Delta^{C-1}$ where $\Delta^{C-1} = \{\boldsymbol{p} \in [0,1]^C : \mathbf{1}^T\boldsymbol{p} = 1, \boldsymbol{p} \geq \mathbf{0}\}$. We want the model to match the most probable class $\boldsymbol{c}_1$ to the true label $y$, where $\boldsymbol{c}(\boldsymbol{x}) \equiv \arg\text{sort}(\boldsymbol{p})$ in descending order. We exchangeably denote the probability value corresponding to the true label $y$ as $p \equiv \boldsymbol{p}_y \in [0, 1]$. The cross-entropy loss, $l = l(\boldsymbol{z}, y) \in \mathbb{R}$, is equivalent to the negative log-likelihood $l = -\log p$. We use the notations $\|\cdot\|$ for the Euclidean $\ell_2$-norm of a vector or for the Euclidean operator norm of a matrix (equivalently, $\|\cdot\|_\sigma$ for a square matrix), $\|\cdot\|_F$ for the Frobenius norm, and $\text{tr}(\cdot)$ for the trace of a (square) matrix.

Starting with a simple computation of the derivatives of the softmax function, Eq (1) (see Appendix A), we can easily derive the following equations in order:

$$\nabla_{\boldsymbol{z}}\boldsymbol{p} = \text{diag}(\boldsymbol{p}) - \boldsymbol{p}\boldsymbol{p}^T \in \mathbb{R}^{C \times C} \tag{1}$$

$$\nabla_{\boldsymbol{z}}p = [\nabla_{\boldsymbol{z}}\boldsymbol{p}]_{:,y} = p(\boldsymbol{e}^y - \boldsymbol{p}) \in \mathbb{R}^C \tag{2}$$

$$\nabla_{\boldsymbol{z}}l = \nabla_{\boldsymbol{z}}p \frac{\partial l}{\partial p} = p(\boldsymbol{e}^y - \boldsymbol{p}) \cdot -\frac{1}{p} = \boldsymbol{p} - \boldsymbol{e}^y \in \mathbb{R}^C \tag{3}$$

$$\nabla_{\boldsymbol{z}}^2 l = \nabla_{\boldsymbol{z}}(\nabla_{\boldsymbol{z}}l) = \nabla_{\boldsymbol{z}}(\boldsymbol{p} - \boldsymbol{e}^y) = \text{diag}(\boldsymbol{p}) - \boldsymbol{p}\boldsymbol{p}^T \in \mathbb{R}^{C \times C} \tag{4}$$

where $\text{diag}(\boldsymbol{p}) = (\delta_{ij}\boldsymbol{p}_i)_{ij} \in \mathbb{R}^{C \times C}$ is a diagonal matrix with $\boldsymbol{p}$ as its diagonal entries, and $\boldsymbol{e}^i = (\delta_{ij})_j \in \mathbb{R}^C$ is a one-hot vector with $i$-th element as 1.

Next, the Hessian of the loss function $l$ for given example $\boldsymbol{x}$ with respect to the model parameter can be expressed as follows:

$$\nabla_{\boldsymbol{\theta}}^2 l = \nabla_{\boldsymbol{\theta}}\boldsymbol{z}\nabla_{\boldsymbol{z}}^2 l\nabla_{\boldsymbol{\theta}}\boldsymbol{z}^T + \sum_{j=1}^{C}\nabla_{\boldsymbol{\theta}}^2 z_j\nabla_{z_j}l \approx \nabla_{\boldsymbol{\theta}}\boldsymbol{z}\nabla_{\boldsymbol{z}}^2 l\nabla_{\boldsymbol{\theta}}\boldsymbol{z}^T \in \mathbb{R}^{m \times m} \tag{5}$$

using a well-known Gauss-Newton approximation. (see, for example, Schraudolph (2002)).

Now, we are ready to consider the training loss for the training set $\mathcal{D}$. We compute the total training loss over $\mathcal{D}$ as $L = \langle l \rangle$ which yields $\nabla L = \langle \nabla l \rangle$ and $\nabla^2 L = \langle \nabla^2 l \rangle$ where $\langle \cdot \rangle$ is the expectation over

the empirical measure of the training set $\mathcal{D}$, equivalently say $\hat{\mathbb{E}}_{\mathcal{D}}[\cdot]$. We use the notation $\langle \cdot \rangle_{\mathcal{S}}$ when averaging over a subset $\mathcal{S}$. Following from Eq (4) and Eq (5), we define the Hessian matrix $\boldsymbol{H}$ for the total loss and its Gauss-Newton approximation matrix $\boldsymbol{G}$ with the matrices $\boldsymbol{M}$ and $\boldsymbol{J}$ as follows:

**Definition 1.** *We call $\boldsymbol{M}$ the logit Hessian, $\boldsymbol{J}$ the Jacobian (of the logit function with respect to the model parameter), $\boldsymbol{H}$ the Hessian, and $\boldsymbol{G}$ the Gauss-Newton approximation defined as follows:*

$$\boldsymbol{M} \equiv \nabla_{\boldsymbol{z}}^2 l = diag(\boldsymbol{p}) - \boldsymbol{p}\boldsymbol{p}^T \in \mathbb{R}^{C \times C} \tag{6}$$

$$\boldsymbol{J}(= \boldsymbol{J}_{\boldsymbol{\theta}}^{\boldsymbol{z}}) \equiv \nabla_{\boldsymbol{\theta}} \boldsymbol{z} \in \mathbb{R}^{m \times C} \tag{7}$$

$$\boldsymbol{H} \equiv \langle \nabla_{\boldsymbol{\theta}}^2 l \rangle \approx \langle \boldsymbol{J}\boldsymbol{M}\boldsymbol{J}^T \rangle \equiv \boldsymbol{G} \in \mathbb{R}^{m \times m} \tag{8}$$

It is interesting to note that while $l$ is dependent on the true label $y$, the logit Hessian $\boldsymbol{M} = \nabla_{\boldsymbol{z}}^2 l$ is independent of $y$, and so are $\boldsymbol{J}$, $\boldsymbol{J}\boldsymbol{M}\boldsymbol{J}^T$, and $\boldsymbol{G}$. In case of the MSE loss $l = \frac{1}{2}\|\boldsymbol{z} - \boldsymbol{e}^y\|^2$, we have $\boldsymbol{M} = \nabla_{\boldsymbol{z}}^2 l = \boldsymbol{I}$ and $\boldsymbol{G} = \langle \boldsymbol{J}\boldsymbol{J}^T \rangle$. We mainly focus on the usual cross-entropy loss and defer the investigation on the MSE loss to Appendix M. From Eq (8), we will often use the approximation $\|\boldsymbol{H}\|_{\sigma} \approx \|\boldsymbol{G}\|_{\sigma}$ as justified in Sagun et al. (2017); Fort & Ganguli (2019), but this approximation sometimes fails in the later phase of training when the top eigenvalues of the Gauss-Newton matrix is not sufficiently isolated from the bulk near 0 (Papyan, 2018). Thus we mainly focus on the early phase of training.

## 4 DECOMPOSITION OF THE HESSIAN $\boldsymbol{H}$

In the previous section, we introduced the Gauss-Newton approximation $\boldsymbol{G}$ of the Hessian $\boldsymbol{H}$, and decomposition of $\boldsymbol{G}$ with the Jacobian $\boldsymbol{J}$ and the logit Hessian $\boldsymbol{M}$, i.e., $\boldsymbol{G} = \langle \boldsymbol{J}\boldsymbol{M}\boldsymbol{J}^T \rangle$. Now, we focus on the logit Hessian matrix $\boldsymbol{M}$ and its eigendecomposition, estimate the top eigenvalue of $\boldsymbol{M}$ with upper/lower bounds, and explore the evolution of the top eigenvalue during training.

### 4.1 EIGENDECOMPOSITION OF THE LOGIT HESSIAN

The lower-dimensional matrix $\boldsymbol{M} \in \mathbb{R}^{C \times C}$ is simple and fully characterized by only the probability vector $\boldsymbol{p}$ as $\boldsymbol{M} = \text{diag}(\boldsymbol{p}) - \boldsymbol{p}\boldsymbol{p}^T$ in Eq (6), but it turns out to be important for understanding the much higher-dimensional matrix $\boldsymbol{G} \in \mathbb{R}^{m \times m}$ ($C \ll m$). Since $\boldsymbol{M} = \text{diag}(\boldsymbol{p}) - \boldsymbol{p}\boldsymbol{p}^T$ is a rank-one modification of a simple diagonal matrix $\text{diag}(\boldsymbol{p})$, we can obtain the eigenvalues and the eigenvectors from the theory of the rank-one modification of the eigenproblem (see, for example, Bunch et al. (1978); Golub (1973) and (Golub & Van Loan, 2013, Section 8.4.3)). Then, the logit Hessian $\boldsymbol{M}$ can be eigendecomposed as $\boldsymbol{M} = \boldsymbol{Q}\boldsymbol{\Lambda}\boldsymbol{Q}^T = \sum_{i=1}^{C} \lambda^{(i)} \boldsymbol{q}^{(i)} \boldsymbol{q}^{(i)T}$ where $\lambda^{(i)}$ is the $i$-th largest eigenvalue of $\boldsymbol{M}$ and $\boldsymbol{q}^{(i)}$ is its corresponding normalized eigenvector. For simplicity, we also use the same ordered index of $(i) \in [C]$ with parentheses for the probability output $\boldsymbol{p} \in \Delta^{C-1}$, i.e., $\boldsymbol{c}_i = (i)$ and $\boldsymbol{p}_{(1)} \geq \boldsymbol{p}_{(2)} \geq \cdots \geq \boldsymbol{p}_{(C)} \geq 0$, because this ordering is related to the eigenvalues $\{\lambda^{(i)}\}_{i=1}^{C}$ as demonstrated in the following theorem. We defer the proof to Appendix B.

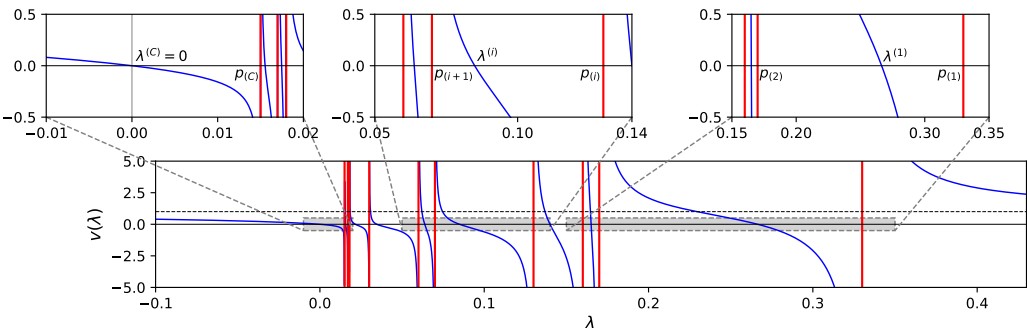

Figure 1: **[Eigenvalues of the logit Hessian] Graph of the secular function $v(\lambda)$ in Eq (9) which has zeros at the eigenvalues $\{\lambda^{(i)}\}_{i=1}^{C}$ of $\boldsymbol{M} = \nabla_{\boldsymbol{z}}^2 l$ (blue curves).** We highlighted the singularities $\lambda = \boldsymbol{p}_{(i)}$ with red vertical lines. It illustrates Theorem 1 (a) and (c).

**Theorem 1** (cf. Golub (1973); Bunch et al. (1978)). *The eigenvalues $\lambda^{(i)}$ ($\lambda^{(1)} \geq \lambda^{(2)} \geq \cdots \geq \lambda^{(C)}$) and the corresponding normalized eigenvectors $\boldsymbol{q}^{(i)}$ of the logit Hessian $\boldsymbol{M} = \nabla_{\boldsymbol{z}}^2 l = diag(\boldsymbol{p}) - \boldsymbol{p}\boldsymbol{p}^T$ satisfy the following properties:*

*(a) The eigenvalue $\lambda^{(i)}$ is the $i$-th largest solution of the following equation:*

$$v(\lambda) = 1 - \sum\nolimits_{i=1}^{C} \frac{\boldsymbol{p}_i^2}{\boldsymbol{p}_i - \lambda} = 0 \tag{9}$$

*(b) The eigenvector $\boldsymbol{q}^{(i)}$ is aligned with the direction of $(diag(\boldsymbol{p}) - \lambda^{(i)}\boldsymbol{I})^{-1}\boldsymbol{p}$*

*(c) $\boldsymbol{p}_{(i+1)} \leq \lambda^{(i)} \leq \boldsymbol{p}_{(i)}$ for $1 \leq i \leq C - 1$, and $\lambda^{(C)} = 0$*

*(d) $\frac{1}{2}Gini(\boldsymbol{p}_{(1)}) \leq \lambda^{(1)} \leq Gini(\boldsymbol{p}_{(1)})$ where $Gini(q) = 1 - q^2 - (1-q)^2 = 2q(1-q)$ is the Gini impurity for the binary case $(q, 1-q)$.*

Figure 1 illustrates the secular function $v(\lambda)$ defined as Eq (9) in Theorem 1 (a). The function $v(\lambda)$ has singularities at the probability values $\{\boldsymbol{p}_{(i)}\}_{i=1}^C$ and zeros at the eigenvalues $\{\lambda^{(i)}\}_{i=1}^C$, satisfying Theorem 1 (c). Moreover, Theorem 1 (c) and (d) provide the upper/lower bounds on the eigenvalues of $\boldsymbol{M}$. Especially, the top eigenvalue $\lambda^{(1)}$ is bounded by $\frac{1}{2}\text{Gini}(\boldsymbol{p}_{(1)}) \leq \lambda^{(1)} \leq \text{Gini}(\boldsymbol{p}_{(1)})$, and thus we call it *impurity*.

## 4.2 EVOLUTION OF IMPURITY

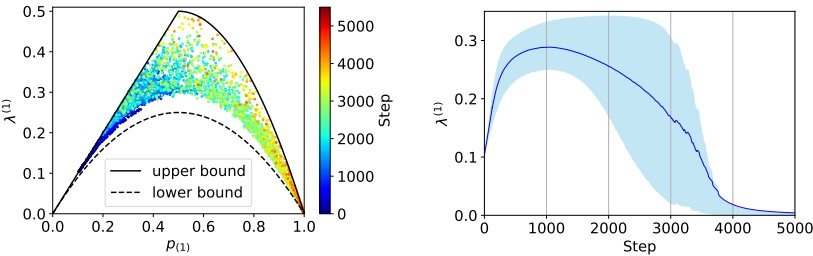

Figure 2: **[Evolution of Impurity] The impurity $\lambda^{(1)}$ increases and then decreases as $\boldsymbol{p}_{(1)}$ increases during training.** Left: The impurity of a typical example is plotted against $\boldsymbol{p}_{(1)}$. We together plot the upper bound $\min\{\boldsymbol{p}_{(1)}, \text{Gini}(\boldsymbol{p}_{(1)})\}$ and the lower bound $\frac{1}{2}\text{Gini}(\boldsymbol{p}_{(1)})$ from Theorem 1 (c) and (d). Right: The impurity is plotted against the training step. Blue curve indicates its mean value $\langle\lambda^{(1)}\rangle$ and sky-blue area shows the 25-75% quantile range of the impurity for the training data.

We explore the top eigenvalue $\lambda^{(1)}$ of $\boldsymbol{M}$ (also referred to as impurity) during training. Figure 2 demonstrates the evolution of the impurity during training, which increases in the beginning and then decreases in the later phase. We trained a model to zero training loss, and thus, for most examples, the probability $\boldsymbol{p}_y$ for the true class $y$ eventually becomes the highest probability $\boldsymbol{p}_{(1)}$. As the top probability $\boldsymbol{p}_{(1)}$ increases from $1/C$ to $1$ during training, the impurity starts from $\lambda^{(1)} \approx \frac{1}{C} \in [\frac{1}{2}\text{Gini}(\frac{1}{C}), \frac{1}{C}] = [\frac{C-1}{C^2}, \frac{1}{C}]$ (Theorem 1 (c) and (d)), increases at the initial phase of the training because it is lower bounded by $\frac{1}{2}\text{Gini}(\boldsymbol{p}_{(1)}) = \boldsymbol{p}_{(1)}(1 - \boldsymbol{p}_{(1)})$ which increases for $\boldsymbol{p}_{(1)} \in [0, 0.5]$. Then $\lambda^{(1)}$ decreases as $\boldsymbol{p}_{(1)}$ becomes larger than $0.5$ which leads $\lambda^{(1)}$ to almost $0$ at the later phase because it is upper bounded by $\text{Gini}(\boldsymbol{p}_{(1)}) = 2\boldsymbol{p}_{(1)}(1 - \boldsymbol{p}_{(1)})$ which decreases for $\boldsymbol{p}_{(1)} \in [0.5, 1]$. Note that Cohen et al. (2021) tried to estimate a similar value, but they use $\boldsymbol{p}_y$, not $\boldsymbol{p}_{(1)}$. We again emphasize that $\boldsymbol{M}$ is independent of the label $y$ but dependent only on the probability output $\boldsymbol{p}$.

## 5 IMPLICIT JACOBIAN REGULARIZATION

In this section, from the results of the previous sections, we aim to derive a relation between the sharpness, the impurity and the Jacobian, and answer how the sharpness of the loss landscape

influences the learning dynamics and the generalization performance. Detailed experimental settings for each Figure are described in Appendix E.

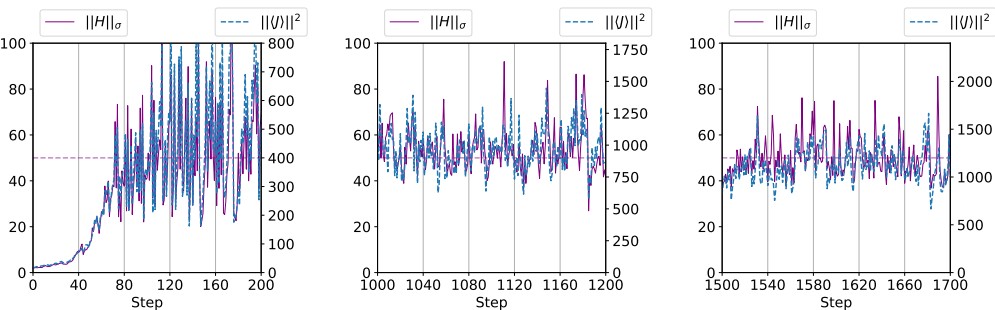

Figure 3: The sharpness $\|\boldsymbol{H}\|_\sigma$ and the Jacobian norm $\|\langle\boldsymbol{J}\rangle\|^2$ show similar oscillating behavior up to a factor $\hat{\lambda}^*$ which is locally constant and slowly changes during training (CIFAR-10, $\eta = 0.04$, $|\mathcal{B}| = 128$; left: 0-200, middle: 1000-1200, right: 1500-1700 steps). We highlighted $\|\boldsymbol{H}\|_\sigma = 2/\eta$ with the dashed horizontal line. Note that they have different right $y$-axes.

## 5.1 SHARPNESS-IMPURITY-JACOBIAN RELATION

We first take a closer look at the sharpness of the loss landscape during training and build a relation between the sharpness $\|\boldsymbol{H}\|_\sigma$, the impurity $\lambda^{(1)}$, and the Jacobian $\boldsymbol{J}$. Since the Gauss-Newton matrix $\boldsymbol{G}$ is known to approximate the true Hessian $\boldsymbol{H}$ well, especially for the top eigenspace (Sagun et al., 2017; Fort & Ganguli, 2019; Papyan, 2019), we can write the sharpness $\|\boldsymbol{H}\|_\sigma$ as follows:

$$\|\boldsymbol{H}\|_\sigma \approx \|\boldsymbol{G}\|_\sigma = \|\langle\boldsymbol{J}\boldsymbol{M}\boldsymbol{J}^T\rangle\|_\sigma \tag{10}$$

It implies the impurity $\|\boldsymbol{M}\|_\sigma$ and the squared norm of the Jacobian $\boldsymbol{J}$ are highly correlated with the sharpness $\|\boldsymbol{H}\|_\sigma$ as demonstrated in the following theorem. We defer the proof to Appendix C.

**Theorem 2.** *For some $0 \le \lambda^* \le \lambda^{(1)}$ for each $\boldsymbol{x} \in \mathcal{D}$, we can bound*

$$\|\langle\boldsymbol{J}\boldsymbol{M}\boldsymbol{J}^T\rangle\|_\sigma = \langle\lambda^*\|\boldsymbol{J}\|^2\rangle \le \langle\lambda^{(1)}\|\boldsymbol{J}\|^2\rangle \tag{11}$$

In other words, $\lambda^*\|\boldsymbol{J}\|^2$ has the expected value of approximately $\|\boldsymbol{H}\|_\sigma$, and we briefly state this relation with $\lambda^*\|\boldsymbol{J}\|^2 \sim \|\boldsymbol{H}\|_\sigma$. Here, $\lambda^*$ in $\lambda^*\|\boldsymbol{J}\|^2$ acts as an adaptive regularization weight for the regularization on the Jacobian norm $\|\boldsymbol{J}\|^2$ as detailed in the following section. Since it is computationally inefficient to track $\|\boldsymbol{J}\|^2$ for every $\boldsymbol{x} \in \mathcal{D}$, we instead investigate $\|\langle\boldsymbol{J}\rangle\|^2$. We expect $\|\boldsymbol{H}\|_\sigma = \hat{\lambda}^*\|\langle\boldsymbol{J}\rangle\|^2$ for some $\hat{\lambda}^*$. In Figure 3, we observe that $\|\boldsymbol{H}\|_\sigma$ and $\|\langle\boldsymbol{J}\rangle\|^2$ show similar oscillating behavior up to a factor $\hat{\lambda}^*$ which is locally constant and slowly changes during training.

## 5.2 IMPLICIT JACOBIAN REGULARIZATION

Now, we are ready to answer how the sharpness of the loss landscape influences the learning dynamics and the generalization performance.

**Growing Jacobian and sharpness in the early phase of training** The weight norm $\|\boldsymbol{\theta}\|$ increases in order to increase the logit norm $\|\boldsymbol{z}\|$ and to minimize the cross-entropy loss during training (Soudry et al., 2018) (see Appendix F for details). This also leads to the increase in the layerwise weight norms and the Jacobian norm. Thus, the increase of the sharpness in the early phase of training can be mainly attributed to the increase of the Jacobian norm $\|\boldsymbol{J}\|$. However, as will be discussed in the following paragraphs, the Jacobian norm does not continuously increase throughout training.

**Oscillatory catapult and the plateau of the sharpness** As the sharpness increases in the beginning, the width of the valley of the loss landscape becomes narrower than the discrete step size of the gradient-based optimization. After the sharpness reaching this threshold, the iterate starts to bounce

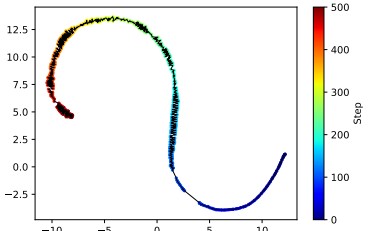 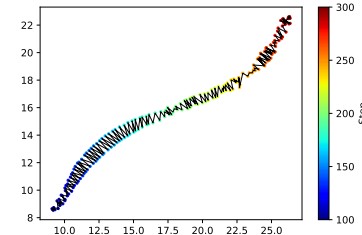

Figure 4: **Oscillatory catapult in the optimization trajectory $\{\theta^{(t)}\}$ (from blue to red) of full-batch GD (McInnes et al., 2018)**. Left: UMAP of the model parameters trained on CIFAR-10 for the first 500 steps. Right: Zoom-in for the oscillatory steps [100, 300]. After few steps ($\sim$100), the sharpness reaches a threshold and the iterate shows the oscillatory behavior with iterative catapult.

off from one side of the valley to the other, and repeats this (Xing et al., 2018; Jastrzębski et al., 2019). Figure 4 shows this oscillatory behavior with iterative catapult after the sharpness reaching the threshold, using UMAP (McInnes et al., 2018). Due to the catapult, the iterate cannot stay in a sharper area and is catapulted to another area. This mechanism is called the catapult mechanism (Lewkowycz et al., 2020). Figure 3 shows fine-grained patterns that the sharpness oscillates around the threshold by the two conflicting effects: the Jacobian norm tends to increase the sharpness and the catapult mechanism reduces it again when the sharpness is over the threshold. Therefore, we can observe the plateau of the sharpness in a coarser scale (see Figure 5). For GD with the MSE loss, the threshold of the sharpness can be easily obtained to be $2/\eta$ ($\eta$ = learing rate), and the threshold value is similar for the cross-entropy loss as shown in Figure 3 (Cohen et al., 2021). The plateau of the sharpness is also observed in the case of SGD, but with a different threshold value depending on the batch size (the smaller the batch size, the lower the plateau) as shown in Figure 6 (Right). This oscillatory catapult and the plateau of the sharpness are attributed to the discrete dynamics of the gradient-based optimization with a finite learning rate and cannot be described by a continuous gradient flow.

**Implicit Jacobian Regularization (IJR)**     Due to this catapult effect, the Jacobian norm cannot continue to increase. In other words, the gradient-based optimization implicitly penalizes the Jacobian norm since $\|\boldsymbol{J}\|^2 \sim \|\boldsymbol{H}\|_\sigma/\lambda^* \lessapprox \frac{2}{\eta\lambda^*}$. This IJR effect begins after the sharpness reaches the threshold. And the regularization effect diminishes as $\lambda^{(1)}$ decreases (the upper limit $\frac{2}{\eta\lambda^*}$ is loosened) with increasing $\boldsymbol{p}_{(1)} \geq 0.5$ in the later phase (see Figure 2, 5 and 6) because $\lambda^* \leq \lambda^{(1)}$ acts as a regularization coefficient. This explains why the behavior of the sharpness in the early phase of training seems to impact the final generalization. Moreover, as $\lambda^{(1)}$ decreases to a very small value, we can observe the decrease in the sharpness. However, the test accuracy is saturated, which provides another counter-example that flat minima generalizes poorly (Dinh et al., 2017).

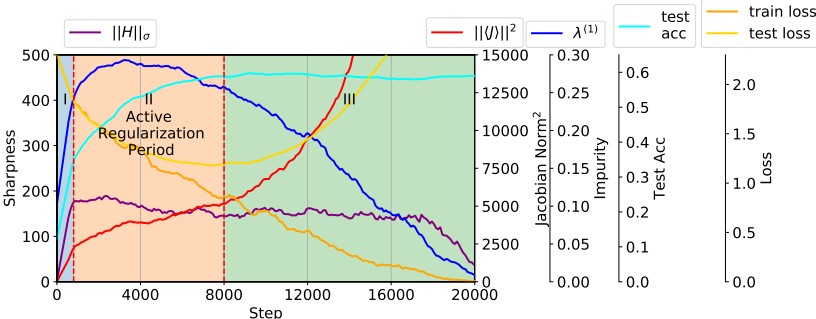

Figure 5: **Three phases of Implicit Jacobian Regularization (IJR).** It shows the IJR effects in the Active Regularization Period (II).

Table 1: **Explicit Jacobian Regularization (EJR) enhances the test accuracy in various settings.** We report improvement ($\Delta$**Acc.**) and Error Reduction Rate (**ERR**) on CIFAR-10/CIFAR-100 when trained with EJR (**+EJR**), compared to the standard training (**Baseline**).

| Dataset | Network Architecture | Batch Size | lr | Regularization Coefficient | Test Accuracy | | $\Delta$Acc. (%p) | ERR (%) |
|---|---|---|---|---|---|---|---|---|
| | | | | | Baseline | +EJR | | |
| CIFAR-10 | SimpleCNN | 128 | 0.003 | $\lambda_{reg} = 0.01$ | 66.71 | **75.40** | +8.69 | 26.10 |
| | | | 0.01 | | 67.88 | **75.62** | +7.74 | 24.10 |
| | | | 0.03 | | 69.83 | **75.53** | +5.70 | 18.89 |
| | | | 0.1 | | 70.33 | **74.29** | +3.96 | 13.35 |
| | | | 0.3 | | 69.34 | **73.47** | +4.13 | 13.47 |
| | | 50000 (full-batch) | 0.01 | $\lambda_{reg} = 0.001$ | 66.81 | **74.43** | +7.62 | 22.96 |
| | | | 0.03 | | 67.72 | **74.31** | +6.59 | 20.42 |
| | | | 0.1 | | 67.53 | **73.69** | +6.16 | 18.97 |
| | | | 0.3 | | 61.08 | **72.15** | +11.07 | 28.44 |
| | WRN-28-10 | 128 | 0.1 | $\rho_{reg} = 2$ | $96.10_{\pm0.05}$ | **97.07** | +0.97 | 24.87 |
| | | | | $\rho_{reg} = 2, \mu_{reg} = 0.03$ | | **97.38** | +1.28 | 32.82 |
| CIFAR-100 | WRN-28-10 | 128 | 0.1 | $\rho_{reg} = 5$ | $80.69_{\pm0.21}$ | **83.73** | +3.04 | 15.74 |

Figure 5 shows that the gradient-based optimization has implicit regularization effects on the Jacobian norm in a certain period. To be specific, there are three phases that the Jacobian norm is (I) initially rapidly increasing before the sharpness reaches near the threshold, (II) actively regularized with a gentle slope, and (III) again exponentially increasing as the regularization effect diminishes (as the regularization weight $\lambda^* \leq \lambda^{(1)}$ decreases) with the slope being gradually steeper. We call the second phase *Active Regularization Period*.

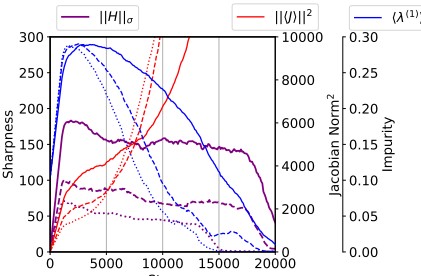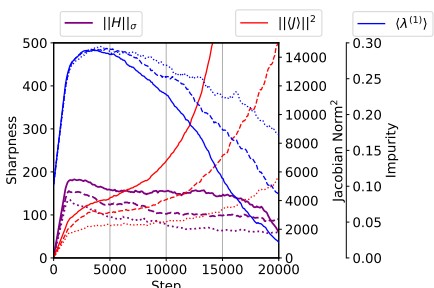

Figure 6: **The IJR effects vary depending on the hyperparameters used in the training.** The plateau of the sharpness and the n-shaped evolution of the impurity are clearly observed as expected. We used SGD (Left) with fixed batch size 128 and different learning rates $\eta = 0.01/0.02/0.03$ and (Right) with fixed $\eta = 0.01$ and different batch sizes 128/64/32 (solid/dashed/dotted lines) on CIFAR-10. Training with a large learning rate and a small batch size (dotted line) penalizes the Jacobian norm more strongly with lower limits of the sharpness. Curves are smoothed for visual clarity.

The evolution of the sharpness of the loss landscape is highly affected by the hyperparameters used in training such as learning rate and batch size (Jastrzębski et al., 2019; Lewkowycz et al., 2020; Cohen et al., 2021). As expected, GD with a large learning rate $\eta$ limits the sharpness with a lower value of $2/\eta$. The large learning rate encourages the stronger implicit regularization on the Jacobian norm. Figure 6 shows, comparing the three red lines (solid/dashed/dotted) with different learning rates (0.01/0.02/0.03) and batch sizes (128/64/32), training with a larger learning rate and a smaller batch size limits the Jacobian norm with a smaller value in the Active Regularization Period. This could explain why training with a large learning rate and a small batch size yields better generalization.

**Explicit Jacobian Regularization (EJR)** To further investigate and boost the effectiveness of IJR, we aim to explicitly regularize the Jacobian norm. However, it is computationally hard to back-propagate through the computation graph of the operator norm of $\|\langle \boldsymbol{J} \rangle\|^2$ for a practical neural network even with a simple iterative method (see Algorithm 2 in Appendix E). Thus, we instead penalize an upper bound, the Frobenius norm $\|\langle \boldsymbol{J} \rangle\|_F^2 (\geq \|\langle \boldsymbol{J} \rangle\|^2)$, with the regularization coefficient $\lambda_{reg}/C$, i.e., we minimize $L + \lambda_{reg}\|\langle \boldsymbol{J} \rangle\|_F^2/C$. See Appendix L for its variants with other regular-

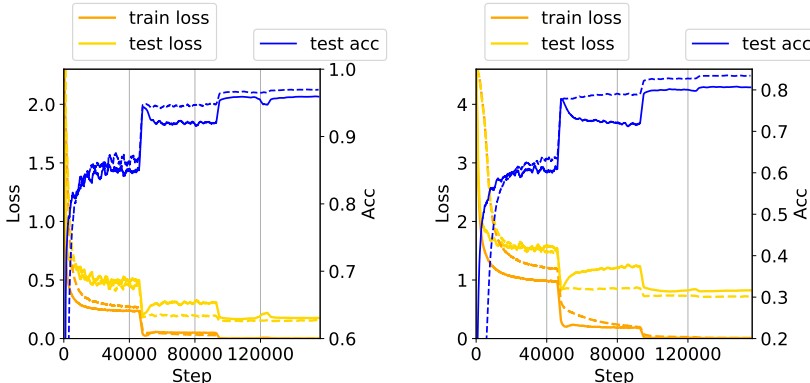

Figure 7: The effectiveness of **EJR** (dashed lines) compared to **Baseline** (solid lines) during training (Left: CIFAR-10, Right: CIFAR-100) on WRN-28-10. With EJR, it mitigates the overfitting, especially after each learning rate decay (undesirable decrease/increase of test accuracy/loss).

ization coefficients $\rho_{reg}$ and $\mu_{reg}$. The Frobenius regularization term can be efficiently computed with an unbiased estimator $\|\langle \boldsymbol{J} \rangle\|_F^2 = C\mathbb{E}_{\boldsymbol{u} \sim U(\mathbb{S}^{C-1})}[\|\langle \boldsymbol{J} \rangle \boldsymbol{u}\|^2] = C\mathbb{E}_{\boldsymbol{u} \sim U(\mathbb{S}^{C-1})}[\|\nabla_{\boldsymbol{\theta}} \langle \boldsymbol{u}^T \boldsymbol{z} \rangle\|^2]$ where $\boldsymbol{u}$ is randomly drawn from the unit hypersphere $\mathbb{S}^{C-1}$. Since the batch-size is large enough, we efficiently use a single sample $\boldsymbol{u} \sim U(\mathbb{S}^{C-1})$ for each batch as suggested in Hoffman et al. (2019). We expect improvements in the generalization performance when introducing EJR. This would support the effectiveness of IJR that it efficiently controls the capacity of the model. Table 1 shows clear improvements in the test accuracy when introducing EJR.

**Connections between the Jacobian and Fisher/Gradient Penalty** Our explanation of the implicit bias in SGD may extend to the catastrophic Fisher explosion (Jastrzebski et al., 2021) with $\boldsymbol{G}$ instead of the Fisher Information Matrix (FIM). The trace of $\boldsymbol{G}$ can be written as follows:

$$\text{tr}(\boldsymbol{G}) = \langle \text{tr}(\boldsymbol{J}\boldsymbol{M}\boldsymbol{J}^T) \rangle = \Big\langle \sum\nolimits_{i=1}^{C-1} \lambda^{(i)} \|\boldsymbol{J}\boldsymbol{q}^{(i)}\|^2 \Big\rangle \approx \langle \text{tr}(\boldsymbol{M})\|\boldsymbol{J}\|_F^2 \rangle / C \qquad (12)$$

where we assume $\|\boldsymbol{J}\boldsymbol{q}^{(i)}\|^2 \approx \|\boldsymbol{J}\|_F^2/C$ since $\sum_{i=1}^{C} \|\boldsymbol{J}\boldsymbol{q}^{(i)}\|^2 = \|\boldsymbol{J}\|_F^2$ (see Figure 11 (Left) in Appendix D for the empirical evidence). Here, the trace of the logit Hessian $\boldsymbol{M}$ can be equivalently written as a $C$-class Gini impurity: $\text{tr}(\boldsymbol{M}) = \sum_{i=1}^{C} \boldsymbol{p}_i(1 - \boldsymbol{p}_i) = 1 - \sum_{i=1}^{C} \boldsymbol{p}_i^2 \equiv \text{Gini}^C(\boldsymbol{p})$ which is $\frac{C-1}{C}$ for the initial uniform distribution and 0 for a one-hot probability. Thus, penalizing $\text{tr}(\boldsymbol{G})$ induces the effects of penalizing $\|\boldsymbol{J}\|_F$, especially in the early phase of training with large $\text{Gini}^C(\boldsymbol{p})$. Thus, as Jastrzebski et al. (2021) argued that Fisher Penalty on the trace of the FIM improves the generalization performance by limiting the memorization, the Jacobian regularization may have similar effects. Moreover, since $\nabla_{\boldsymbol{\theta}} l(\boldsymbol{z}, \hat{y}) = \nabla_{\boldsymbol{\theta}} \boldsymbol{z} \nabla_{\boldsymbol{z}} l(\boldsymbol{z}, \hat{y}) = \boldsymbol{J}(\boldsymbol{p} - \boldsymbol{e}^{\hat{y}})$, the trace of the FIM they approximately used is simply $\|\hat{\mathbb{E}}_{\boldsymbol{x} \sim \mathcal{B}} \mathbb{E}_{\hat{y} \sim \boldsymbol{p}}[\nabla_{\boldsymbol{\theta}} l(\boldsymbol{z}, \hat{y})]\|^2 = \|\hat{\mathbb{E}}_{\boldsymbol{x} \sim \mathcal{B}} \mathbb{E}_{\hat{y} \sim \boldsymbol{p}}[\boldsymbol{J}(\boldsymbol{p} - \boldsymbol{e}^{\hat{y}})]\|^2$ with a single sample $\hat{y}$, the gradient norm penalty (Barrett & Dherin, 2021) is $\|\hat{\mathbb{E}}_{(\boldsymbol{x},y) \sim \mathcal{B}}[\boldsymbol{J}(\boldsymbol{p} - \boldsymbol{e}^y)]\|$ and the explicit Jacobian regularizer is $\|\boldsymbol{J}\|_F^2 = C\mathbb{E}_{\boldsymbol{u} \sim U(\mathcal{S}^{C-1})} \|\hat{\mathbb{E}}_{(\boldsymbol{x},y) \sim \mathcal{B}}[\boldsymbol{J}\boldsymbol{u}]\|^2$. In each case, we emphasize that the Jacobian $\boldsymbol{J}$ plays an important role in the generalization performance.

## 6 CONCLUSION

We have investigated the eigensystem of the Hessian through a new decomposition using eigen-decomposition of the low-dimensional logit Hessian. By doing so, we could provide a simple and intuitive explanation on the relation between the gradient-based optimization, the learning dynamics and the generalization performance of neural networks. We hope this research could help answer other intriguing questions on the learning dynamics and generalization.

ETHICS STATEMENT

No ethics statement is needed for this study.

REPRODUCIBILITY STATEMENT

We provide code to reproduce our experiments. We refer the readers to the supplementary material (README.md).

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

# A   PROOF OF EQ (1)

$$\nabla_{\boldsymbol{z}}\boldsymbol{p} = \mathrm{diag}(\boldsymbol{p}) - \boldsymbol{p}\boldsymbol{p}^T \in \mathbb{R}^{C \times C} \tag{1}$$

*Proof.* By definition of the softmax function,

$$\boldsymbol{p}_j = [\mathrm{softmax}(\boldsymbol{z})]_j = \frac{\exp(\boldsymbol{z}_j)}{\sum_k \exp(\boldsymbol{z}_k)} = \exp(\boldsymbol{z}_j)s^{-1} \tag{13}$$

where $s = \sum_k \exp(\boldsymbol{z}_k)$, we have

$$\nabla_{\boldsymbol{z}_i}\boldsymbol{p}_j = \begin{cases} -\exp(\boldsymbol{z}_j)s^{-2}\exp(\boldsymbol{z}_i) = -\boldsymbol{p}_i\boldsymbol{p}_j, & \text{if } i \neq j \\ -\exp(\boldsymbol{z}_i)s^{-2}\exp(\boldsymbol{z}_i) + \exp(\boldsymbol{z}_i)s^{-1} = -\boldsymbol{p}_i^2 + \boldsymbol{p}_i, & \text{if } i = j \end{cases} \tag{14}$$

which leads to $\nabla_{\boldsymbol{z}}\boldsymbol{p} = (\nabla_{\boldsymbol{z}_i}\boldsymbol{p}_j)_{ij} = -\boldsymbol{p}\boldsymbol{p}^T + \mathrm{diag}(\boldsymbol{p})$. $\qquad\square$

# B   PROOF OF THEOREM 1

**Theorem** (restated). *The eigenvalues $\lambda^{(i)}$ ($\lambda^{(1)} \geq \lambda^{(2)} \geq \cdots \geq \lambda^{(C)}$) and the corresponding normalized eigenvectors $\boldsymbol{q}^{(i)}$ of the logit Hessian $\boldsymbol{M} = \nabla_{\boldsymbol{z}}^2 l = diag(\boldsymbol{p}) - \boldsymbol{p}\boldsymbol{p}^T$ satisfy the following properties:*

(a) *The eigenvalue $\lambda^{(i)}$ is the $i$-th largest solution of the following equation:*

$$v(\lambda) = 1 - \sum_{i=1}^{C} \frac{\boldsymbol{p}_i^2}{\boldsymbol{p}_i - \lambda} = 0 \tag{9}$$

(b) *The eigenvector $\boldsymbol{q}^{(i)}$ is aligned with the direction of $(diag(\boldsymbol{p}) - \lambda^{(i)}\boldsymbol{I})^{-1}\boldsymbol{p}$*

(c) $\boldsymbol{p}_{(i+1)} \leq \lambda^{(i)} \leq \boldsymbol{p}_{(i)}$ *for $1 \leq i \leq C - 1$, and $\lambda^{(C)} = 0$*

(d) $\frac{1}{2}Gini(\boldsymbol{p}_{(1)}) \leq \lambda^{(1)} \leq Gini(\boldsymbol{p}_{(1)})$ *where $Gini(q) = 1 - q^2 - (1-q)^2 = 2q(1-q)$ is the Gini impurity for the binary case $(q, 1-q)$.*

*Proof.* The eigenvalues $\lambda^{(i)}$ of $\boldsymbol{M} = \mathrm{diag}(\boldsymbol{p}) - \boldsymbol{p}\boldsymbol{p}^T$ are the zeros of the following characteristic polynomial:

$$\phi_{\boldsymbol{M}}(\lambda) = \det(\mathrm{diag}(\boldsymbol{p}) - \boldsymbol{p}\boldsymbol{p}^T - \lambda\boldsymbol{I}) \tag{15}$$

$$= \det(\mathrm{diag}(\boldsymbol{p}) - \lambda\boldsymbol{I})\det(\boldsymbol{I} - (\mathrm{diag}(\boldsymbol{p}) - \lambda\boldsymbol{I})^{-1}\boldsymbol{p}\boldsymbol{p}^T) \tag{16}$$

$$= \prod_{i=1}^{C}(\boldsymbol{p}_i - \lambda)\left(1 - \sum_{j=1}^{C} \frac{\boldsymbol{p}_j^2}{\boldsymbol{p}_j - \lambda}\right) \tag{17}$$

where the second equality follows from $\boldsymbol{A} - \boldsymbol{p}\boldsymbol{p}^T = \boldsymbol{A}(\boldsymbol{I} - \boldsymbol{A}^{-1}\boldsymbol{p}\boldsymbol{p}^T)$ with the matrix $\boldsymbol{A} = \mathrm{diag}(\boldsymbol{p}) - \lambda\boldsymbol{I}$, and the third inequality holds because $\det(\boldsymbol{I} + \boldsymbol{u}\boldsymbol{v}^T) = 1 + \boldsymbol{u}^T\boldsymbol{v}$ for vectors $\boldsymbol{u}$ and $\boldsymbol{v}$. Then it is equivalent to solving the following equation:

$$v(\lambda) = 1 - \sum_{i=1}^{C} \frac{\boldsymbol{p}_i^2}{\boldsymbol{p}_i - \lambda} = 0 \tag{18}$$

which implies (a). Note that this result also implies (c) as shown in Figure 1.

Next, to prove (b), put $\boldsymbol{A} \equiv \mathrm{diag}(\boldsymbol{p}) - \lambda\boldsymbol{I}$ and $\boldsymbol{q} \equiv \boldsymbol{A}^{-1}\boldsymbol{p}$. Then it is required to show that $(\boldsymbol{M} - \lambda\boldsymbol{I})\boldsymbol{q} = (\boldsymbol{A} - \boldsymbol{p}\boldsymbol{p}^T)\boldsymbol{q} = 0$ for the eigenvalues $\lambda = \lambda^{(i)}$. We have

$$(\boldsymbol{A} - \boldsymbol{p}\boldsymbol{p}^T)\boldsymbol{q} = (\boldsymbol{A} - \boldsymbol{p}\boldsymbol{p}^T)\boldsymbol{A}^{-1}\boldsymbol{p} = \boldsymbol{p} - \boldsymbol{p}\boldsymbol{p}^T\boldsymbol{A}^{-1}\boldsymbol{p} \tag{19}$$

Here, $(\boldsymbol{p}\boldsymbol{p}^T\boldsymbol{A}^{-1}\boldsymbol{p})_i = \sum_{j,k}\boldsymbol{p}_i\boldsymbol{p}_j\boldsymbol{A}_{jk}^{-1}\boldsymbol{p}_k = \sum_{j,k}\boldsymbol{p}_i\boldsymbol{p}_j\delta_{jk}(\boldsymbol{p}_j - \lambda)^{-1}\boldsymbol{p}_k = \sum_k\boldsymbol{p}_i\boldsymbol{p}_k(\boldsymbol{p}_k - \lambda)^{-1}\boldsymbol{p}_k = \boldsymbol{p}_i\sum_k\boldsymbol{p}_k^2/(\boldsymbol{p}_k - \lambda) = \boldsymbol{p}_i$. The last equality holds for the eigenvalues $\lambda = \lambda^{(i)}$ which follows from (a).

Now, we want to prove the statement (c). Since

$$\lambda^{(i)}(\boldsymbol{C}) \le \lambda^{(j)}(\boldsymbol{A}) + \lambda^{(k)}(\boldsymbol{B}) \text{ if } k + j - i = 1 \tag{20}$$

$$\lambda^{(i)}(\boldsymbol{C}) \ge \lambda^{(j)}(\boldsymbol{A}) + \lambda^{(k)}(\boldsymbol{B}) \text{ if } k + j - i = C \tag{21}$$

for $\boldsymbol{C} = \boldsymbol{A} + \boldsymbol{B} \in \mathbb{R}^{C \times C}$ where $\lambda^{(i)}(\boldsymbol{D})$ is the $i$-th largest eigenvalue of a matrix $\boldsymbol{D}$ (Weyl, 1912; Fulton, 2000), we can get $\lambda^{(i)}(\boldsymbol{C}) \le \lambda^{(i)}(\boldsymbol{A}) + \lambda^{(1)}(\boldsymbol{B})$ and $\lambda^{(i)}(\boldsymbol{C}) \ge \lambda^{(i+1)}(\boldsymbol{A}) + \lambda^{(C-1)}(\boldsymbol{B})$. Thus, for $\boldsymbol{A} = \text{diag}(\boldsymbol{p})$ and $\boldsymbol{B} = -\boldsymbol{p}\boldsymbol{p}^T$, we can get

$$\boldsymbol{p}_{(i+1)} \le \lambda^{(i)}(\boldsymbol{M}) \le \boldsymbol{p}_{(i)} \text{ for } 1 \le i \le C - 1 \tag{22}$$

since $\lambda^{(i)}(\boldsymbol{A}) = \boldsymbol{p}_{(i)}$, $\lambda^{(i+1)}(\boldsymbol{A}) = \boldsymbol{p}_{(i+1)}$ and $\lambda^{(1)}(\boldsymbol{B}) = \lambda^{(C-1)}(\boldsymbol{B}) = 0$. Moreover, since $\boldsymbol{M}\boldsymbol{1} = \boldsymbol{p} - \boldsymbol{p}\boldsymbol{p}^T\boldsymbol{1} = \boldsymbol{p} - \boldsymbol{p}\sum_i\boldsymbol{p}_i = \boldsymbol{0}$, the smallest eigenvalue is $\lambda^{(C)} = 0$.

Lastly, we prove the statement (d). From the Gershgorin circle theorem (Gershgorin, 1931), we have

$$\lambda^{(1)} \in \bigcup_i \overline{\mathbb{B}}(\boldsymbol{M}_{ii}, \sum_{j \ne i} |\boldsymbol{M}_{ij}|) = \overline{\mathbb{B}}(\boldsymbol{p}_{(1)}(1 - \boldsymbol{p}_{(1)}), \boldsymbol{p}_{(1)}(1 - \boldsymbol{p}_{(1)})) = [0, 2\boldsymbol{p}_{(1)}(1 - \boldsymbol{p}_{(1)})] \tag{23}$$

which implies $\lambda^{(1)} \le 2\boldsymbol{p}_{(1)}(1 - \boldsymbol{p}_{(1)})$. Note that $\boldsymbol{p}_{(1)}(1 - \boldsymbol{p}_{(1)}) \ge \boldsymbol{p}_{(i)}(1 - \boldsymbol{p}_{(i)})$ since $g(t) = t(1 - t)$ is increasing for $0 \le t \le 0.5$. In detail, if $\boldsymbol{p}_{(1)} \ge 0.5$, since $\boldsymbol{p}_{(i)} \le 1 - \boldsymbol{p}_{(1)} \le 0.5$, we have $g(\boldsymbol{p}_{(i)}) \le g(1 - \boldsymbol{p}_{(1)}) = g(\boldsymbol{p}_{(1)})$. Otherwise ($\boldsymbol{p}_{(1)} < 0.5$), since $\boldsymbol{p}_{(i)} \le \boldsymbol{p}_{(1)}$, it leads to the same inequality $g(\boldsymbol{p}_{(i)}) \le g(\boldsymbol{p}_{(1)})$. With the Rayleigh principle, we can express the largest eigenvalue as $\lambda^{(1)} = \max_{\|\boldsymbol{u}\|_2 = 1} \boldsymbol{u}^T \boldsymbol{M}\boldsymbol{u}$, and thus $\boldsymbol{e}^{(1)T}\boldsymbol{M}\boldsymbol{e}^{(1)} = \boldsymbol{M}_{(1)(1)} = \boldsymbol{p}_{(1)}(1 - \boldsymbol{p}_{(1)}) \le \lambda^{(1)}$.

$\square$

# C  PROOF OF THEOREM 2

**Theorem** (restated). *For some $0 \le \lambda^* \le \lambda^{(1)}$ for each $\boldsymbol{x} \in \mathcal{D}$, we can bound*

$$\|\langle \boldsymbol{J}\boldsymbol{M}\boldsymbol{J}^T\rangle\|_\sigma = \langle \lambda^* \|\boldsymbol{J}\|^2\rangle \le \langle \lambda^{(1)} \|\boldsymbol{J}\|^2\rangle \tag{11}$$

*Proof.* We start with the Rayleigh principle:

$$\|\langle \boldsymbol{J}\boldsymbol{M}\boldsymbol{J}^T\rangle\|_\sigma = \max_{\|\boldsymbol{q}\|=1} \boldsymbol{q}^T\langle \boldsymbol{J}\boldsymbol{M}\boldsymbol{J}^T\rangle\boldsymbol{q} = \max_{\|\boldsymbol{q}\|=1} \langle \boldsymbol{q}^T\boldsymbol{J}\boldsymbol{M}\boldsymbol{J}^T\boldsymbol{q}\rangle \tag{24}$$

Since $\boldsymbol{M} = \sum_i \lambda^{(i)}\boldsymbol{q}^{(i)}\boldsymbol{q}^{(i)T}$, we can continue by putting $\boldsymbol{v} = \boldsymbol{J}^T\boldsymbol{q}$, and then

$$Eq(24) = \max_{\|\boldsymbol{q}\|=1} \langle \boldsymbol{v}^T\boldsymbol{M}\boldsymbol{v}\rangle = \max_{\|\boldsymbol{q}\|=1} \langle \sum_i \lambda^{(i)}(\boldsymbol{q}^{(i)T}\boldsymbol{v})^2\rangle \tag{25}$$

Then by putting $\tilde{\lambda} = \sum_i \gamma^{(i)}\lambda^{(i)}$ with $\gamma^{(i)} = (\boldsymbol{q}^{(i)T}\boldsymbol{v})^2/\sum_i(\boldsymbol{q}^{(i)T}\boldsymbol{v})^2 \ge 0$ ($\sum_i \gamma_i = 1$),

$$Eq(25) = \max_{\|\boldsymbol{q}\|=1} \langle \tilde{\lambda}\sum_i(\boldsymbol{q}^{(i)T}\boldsymbol{v})^2\rangle = \max_{\|\boldsymbol{q}\|=1} \langle \tilde{\lambda}\sum_i(\boldsymbol{q}^{(i)T}\boldsymbol{J}^T\boldsymbol{q})^2\rangle \tag{26}$$

$$= \max_{\|\boldsymbol{q}\|=1} \langle \tilde{\lambda}\sum_i(\boldsymbol{q}^{(i)T}(\boldsymbol{J}^T\boldsymbol{q}))^2\rangle \tag{27}$$

Since $\{\boldsymbol{q}^{(i)}\}_{i=1}^C$ is an orthonormal basis of $\mathbb{R}^C$ (eigenvectors of a symmetric matrix $\boldsymbol{M}$), we have the following by putting $\lambda^* = \frac{\|\boldsymbol{J}^T\boldsymbol{q}^*\|^2}{\|\boldsymbol{J}\|^2}\tilde{\lambda}$ with $\boldsymbol{q}^* = \arg\max_{\|\boldsymbol{q}\|=1} \langle \tilde{\lambda}\|\boldsymbol{J}^T\boldsymbol{q}\|^2\rangle$,

$$Eq(27) = \max_{\|\boldsymbol{q}\|=1} \langle \tilde{\lambda}\|\boldsymbol{J}^T\boldsymbol{q}\|^2\rangle = \langle \tilde{\lambda}\|\boldsymbol{J}^T\boldsymbol{q}^*\|^2\rangle = \langle \lambda^*\|\boldsymbol{J}\|^2\rangle \tag{28}$$

Because of the definition of $\lambda^*$ and $\tilde{\lambda}$ with $\|q^*\| = 1$ and $\sum_i \gamma^{(i)} = 1$ ($\gamma^{(i)} \geq 0$), we have

$$\lambda^* \leq \tilde{\lambda} \leq \lambda^{(1)} \tag{29}$$

which leads to the final inequality in Eq (11). The equality holds when $\|J^T q^*\| = \|J\|$, $\gamma^{(1)} = 1$ and $\gamma^{(i)} = 0$ ($i \neq 1$) for all $x \in \mathcal{D}$.

$\square$

## D   GRADIENT DESCENT IN THE TOP HESSIAN SUBSPACE

Gur-Ari et al. (2018) showed that the gradient of the loss quickly converges to a tiny subspace spanned by a few top eigenvectors of the Hessian after a short training. Then, the top Hessian subspace does not evolve much, which implies gradient descent happens in a tiny subspace. However, the underlying mechanism has not been fully understood.

**Direction of $q^{(i)}$ and two salient elements**   We investigate the direction of the eigenvector $q^{(i)}$ ($1 \le i \le C - 1$) of $M$. The eigenvector

$$q^{(i)} = \alpha \left( \frac{p_j}{p_j - \lambda^{(i)}} \right)_j \tag{30}$$

can be obtained from Theorem 1 (b) for some $\alpha > 0$. Here, the magnitude of the denominator $|p_j - \lambda^{(i)}|$ is small for the two indices $j = (i), (i+1)$, and is large for the others. This is because the eigenvalue $\lambda^{(i)}$ lies between $p_{(i+1)}$ and $p_{(i)}$ (Theorem 1 (c)). Therefore, the eigenvector $q^{(i)}$ has a relatively large positive value in $q^{(i)}_{(i)}$ and a large negative value in $q^{(i)}_{(i+1)}$ compared to the other components.

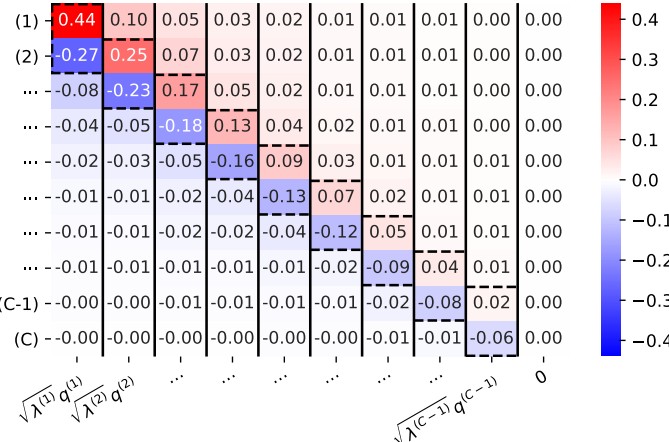

Figure 8: Heatmap of the matrix $Q\Lambda^{1/2} = [\sqrt{\lambda^{(1)}}q^{(1)}; \cdots; \sqrt{\lambda^{(C-1)}}q^{(C-1)}; \mathbf{0}]$ averaged over the training set $\mathcal{D}$ where $M = Q\Lambda Q^T$. Each column of $Q\Lambda^{1/2}$ visualizes the color-encoded direction of $q^{(i)}$ multiplied by $\sqrt{\lambda^{(i)}}$. We highlighted the elements $q^{(i)}_{(i)}$ and $q^{(i)}_{(i+1)}$ with the dashed boxes for $0 \le i \le C - 1$ (see Theorem 1 (b)).

Figure 8 shows the directions of $q^{(i)}$ with the heatmap of the matrix $Q\Lambda^{1/2}$ where $Q\Lambda^{1/2} = [\sqrt{\lambda^{(1)}}q^{(1)}; \cdots; \sqrt{\lambda^{(C-1)}}q^{(C-1)}; \mathbf{0}] \in \mathbb{R}^{C \times C}$ for $M = Q\Lambda Q^T$. As expected, considering each column of $Q\Lambda^{1/2}$, the eigenvector $q^{(i)}$ is colored in red (+) at $q^{(i)}_{(i)}$ and in blue (−) at $q^{(i)}_{(i+1)}$ for $1 \le i \le C - 1$. The two salient elements are highlighted with the dashed boxes.

**Direction of $Jq^{(i)}$ and margin maximization**   In light of the previous discussion, the direction of

$$Jq^{(i)} = (Jq^{(i)})|_{\theta=\theta^{(t)}} = \nabla_\theta \left( q^{(i)}(\theta^{(t)})^T z(\theta) \right) |_{\theta=\theta^{(t)}} \in \mathbb{R}^m \tag{31}$$

is approximately a direction maximizing $q^{(i)}_{(i)} z_{(i)} + q^{(i)}_{(i+1)} z_{(i+1)}$ at the current parameter $\theta^{(t)} \in \Theta$ because the other terms are relatively small. In other words, it tends to maximize the margin $z_{(i)} - z_{(i+1)}$ in the logit space $\mathcal{Z}$ between the two classes $(i)$ and $(i+1)$ (see Figure 8). In particular, $Jq^{(1)}$ is approximately a direction that maximizes the margin between the most likely class and the second most likely class.

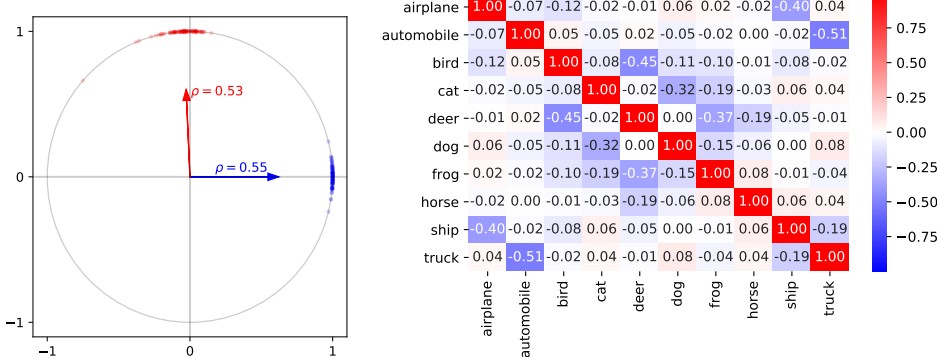

Figure 9: **There are $C$ clusters of $\boldsymbol{m} = \sqrt{\lambda^{(1)}}\boldsymbol{J}\boldsymbol{q}^{(1)}$ according to the most likely class (not the true class). Left: (Within-class similarity)** Directional data of $\boldsymbol{m}$ from $\mathcal{D}_i$ and $\mathcal{D}_j$ for the two classes, dog (blue) and automobile (red). They are projected onto the 2D-plane spanned by the two mean vectors indicated with the arrows. We highlight the MRL $\rho$ for each class. The directional data of $\boldsymbol{m}$ are concentrated within the class but separated from each other. **Right: (Between-class dissimilarity)** Cosine similarities between each pair of $\{\overline{\boldsymbol{m}}^i\}$. They are mostly orthogonal, but some pairs are even negatively aligned, for example, automobile and truck. This is because the examples predicted to be automobile mostly have the second most probable class as truck.

**Clustering of $\boldsymbol{J}\boldsymbol{q}^{(1)}$ and the most probable class**   We first define following subsets of the training set according to the most probable class (and the second most one): $\mathcal{D}_i = \{\boldsymbol{x} \in \mathcal{D} : \boldsymbol{c}_1(\boldsymbol{x}) = i\} \subset \mathcal{D}$ and $\mathcal{D}_{ij} = \{\boldsymbol{x} \in \mathcal{D} : \boldsymbol{c}_1(\boldsymbol{x}) = i, \boldsymbol{c}_2(\boldsymbol{x}) = j\} \subset \mathcal{D}$ for $i \neq j \in [C]$. Note that $\mathcal{D}_i = \bigcup_{j \neq i} \mathcal{D}_{ij}$. Given two examples from $\mathcal{D}_{ij}$, their $\boldsymbol{J}\boldsymbol{q}^{(1)}$ are expected to be highly aligned to each other. This is because the direction of $\boldsymbol{J}\boldsymbol{q}^{(1)}$ is approximately a direction of maximizing the margin and of learning the features to discriminate the class $i$ from the class $j$. Moreover, two examples from $\mathcal{D}_i$ also have highly-aligned $\boldsymbol{J}\boldsymbol{q}^{(1)}$. Figure 9 (Left) shows the concentration of the directional data of $\boldsymbol{J}\boldsymbol{q}^{(1)}$ from $\mathcal{D}_i$. We also compute the mean resultant length (MRL) to measure the concentration. The MRL $\rho$ of the directional variable $V \in \mathbb{S}^{m-1} \equiv \{\boldsymbol{v} \in \mathbb{R}^m : \|\boldsymbol{v}\| = 1\}$ defined as $\rho \equiv \|\mathbb{E}[V]\| \in [0, 1]$ indicates how $V$ is distributed (the higher, the more concentrated).

Now, we focus on $\boldsymbol{m} \equiv \sqrt{\lambda^{(1)}}\boldsymbol{J}\boldsymbol{q}^{(1)}$ as the other $\lambda^{(i)}$-terms are dominated by the $\lambda^{(1)}$-term after a few epochs (see Appendix J for details). Then, we follow a similar approach from (Papyan, 2019) and provide the following equation:

$$\langle \boldsymbol{m}\boldsymbol{m}^T \rangle = \sum_{i=1}^{C} \gamma_i \langle \boldsymbol{m}\boldsymbol{m}^T \rangle_{\mathcal{D}_i} = \sum_{i=1}^{C} \gamma_i (\overline{\boldsymbol{m}}^i \overline{\boldsymbol{m}}^{iT} + \langle (\boldsymbol{m} - \overline{\boldsymbol{m}}^i)(\boldsymbol{m} - \overline{\boldsymbol{m}}^i)^T \rangle_{\mathcal{D}_i}) \quad (32)$$

where $\gamma_i = |\mathcal{D}_i|/|\mathcal{D}|$ and $\overline{\boldsymbol{m}}^i = \langle \boldsymbol{m} \rangle_{\mathcal{D}_i}$. Here, the covariance term $\langle (\boldsymbol{m} - \overline{\boldsymbol{m}}^i)(\boldsymbol{m} - \overline{\boldsymbol{m}}^i)^T \rangle_{\mathcal{D}_i}$ is weak as $\boldsymbol{m}$ is concentrated within $\mathcal{D}_i$, and thus we can roughly approximate $\boldsymbol{G}$ with $\sum_{i=1}^{C} \gamma_i \overline{\boldsymbol{m}}^i \overline{\boldsymbol{m}}^{iT}$. This implies that the top eigensubspace of the Hessian highly overlaps with the at most $C$-dimensional subspace spanned by $\{\overline{\boldsymbol{m}}^i\}_{i=1}^C$. Figure 9 (Right) demonstrates that the mean vectors $\{\overline{\boldsymbol{m}}^i\}_{i=1}^C$ are well separated from each other. This also implies the outliers in the Hessian spectrum (Sagun et al., 2016; 2017).

**Why gradient descent happens mostly in the top Hessian subspace?**   Given input $\boldsymbol{x}$, after the model becomes to correctly predict the true label $y$, the gradient descent direction $-\boldsymbol{g} = \boldsymbol{J}(\boldsymbol{e}^y - \boldsymbol{p})$ used in the training tends to be highly aligned with $\boldsymbol{J}\boldsymbol{q}^{(1)}$. This is because $\boldsymbol{e}^y - \boldsymbol{p}$ and $\boldsymbol{q}^{(1)}$ both have similar direction. They have positive values $1 - \boldsymbol{p}_y$ and $\boldsymbol{q}_{(1)}^{(1)}$ in $y(= (1))$-th element, negative value $-\boldsymbol{p}_i$ and $\boldsymbol{q}_i^{(1)}$ in the others, and especially large negative value for the second most probable class $i = (2)$. Figure 10 (Middle) shows the cosine similarity between the gradient descent direction $-\boldsymbol{g}$ and $\boldsymbol{J}\boldsymbol{q}^{(1)}$. As expected, they are highly aligned with the cosine similarity near 1 as the two vectors $\boldsymbol{e}^y - \boldsymbol{p}$ and $\boldsymbol{q}^{(1)}$ become more aligned to each other.

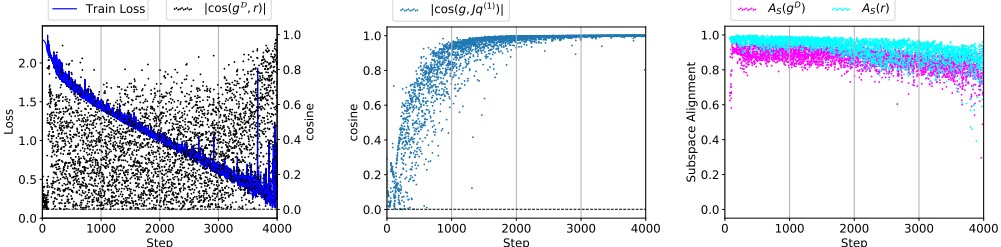

Figure 10: **Left: Total gradient $g^{\mathcal{D}} = \langle g \rangle$ is aligned with the top eigenvector $r$ of the Hessian $H$ at each step during training (Jastrzębski et al., 2019; Gur-Ari et al., 2018).** They have large cosine similarities considering that they are very high-dimensional. We highlighted the cosine value for random $m$-dimensional vectors in $\Theta$ with the dashed horizontal line (about 1e-3). **Middle: $Jq^{(1)}$ (or $m$) is highly aligned with the gradient $g$ for given example at each step during training.** They have cosine similarities near 1 as the model becomes to correctly predict the true label. See Figure 2 (Right) together. **Right: Total gradient $g^{\mathcal{D}}$ and the top eigenvector $r$ of the Hessian $H$ mostly lie in the at most $C$-dimensional subspace $\mathcal{S}$ spanned by $\{\overline{m}^i\}_{i=1}^C$.** The subspace alignment measure $A_{\mathcal{S}}$ is defined in Eq (33).

Next, we move on to the subspace $\mathcal{S} \equiv \mathrm{span}(\{\overline{m}^i\}_{i=1}^C)$ spanned by $\{\overline{m}^i\}_{i=1}^C$. As each $m = \sqrt{\lambda^{(1)}} Jq^{(1)}$ is highly aligned with $-g$, it is reasonably expected that the total gradient $g^{\mathcal{D}}$ lies in the subspace $\mathcal{S}$. To measure how much the vector $v \neq 0$ is aligned with the subspace $\mathcal{S}$, we define the cosine similarity between the vector $v$ and its projection $P_{\mathcal{S}}(v)$ onto the subspace $\mathcal{S}$ as follows:

$$A_{\mathcal{S}}(v) \equiv \cos(v, P_{\mathcal{S}}(v)) = \|P_{\mathcal{S}}(v)\| / \|v\| \tag{33}$$

In particular, $A_{\mathcal{S}}(v) = 0$ means $v \in \mathcal{S}^{\perp}$ and $A_{\mathcal{S}}(v) = 1$ means $v \in \mathcal{S}$. Figure 10 (Right) shows the high alignment of the total gradient $g^{\mathcal{D}}$ in the subspace $\mathcal{S} \equiv \mathrm{span}(\{\overline{m}^i\}_{i=1}^C)$ although the subspace $\mathcal{S}$ is of dimension at most $C \ll m$. Moreover, since $G$ can be roughly approximated by $\sum_{i=1}^C \gamma^i \overline{m}^i \overline{m}^{iT}$, the top eigenvector $r$ of the Hessian $H$ is also highly aligned with the subspace $\mathcal{S}$.

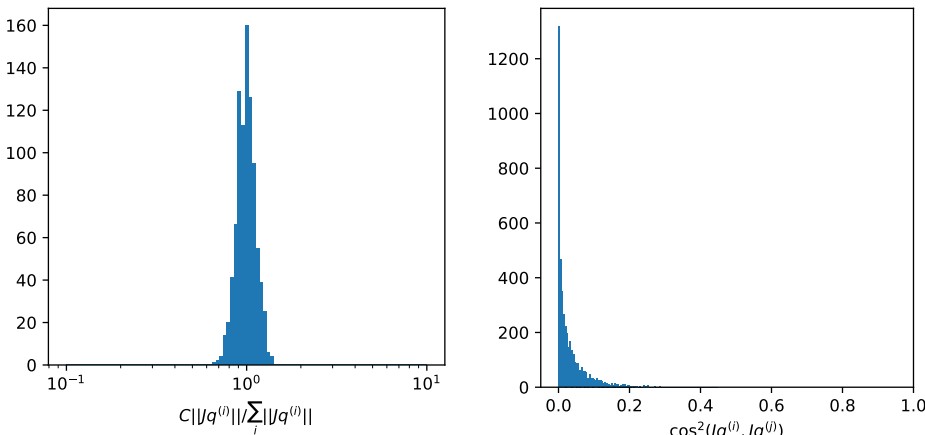

Figure 11: **Histograms of $\dfrac{\|Jq^{(i)}\|}{\sum_j \|Jq^{(j)}\|/C}$ and $\cos^2(Jq^{(i)}, Jq^{(j)})$.**

# E  EXPERIMENTAL SETTINGS

## E.1  DATA

We use the CIFAR-10 dataset (Krizhevsky et al. (2009), `https://www.cs.toronto.edu/~kriz/cifar.html`) and the MNIST dataset which have $C = 10$ number of classes. We also conduct some experiments on the CIFAR-100 dataset with the number of classes $C = 100$. We mostly do not use the data augmentation for training (1) not to introduce the randomness in the training loss and (2) to allow the training loss to converge to a small value. In case of the WRN-28-10 (Zagoruyko & Komodakis, 2016), we use the data augmentation to improve the performance.

## E.2  NETWORK ARCHITECTURES

We use the following models: VGG-11 (VGG) (Simonyan & Zisserman, 2015) without batch-normalization, VGG for CIFAR-100 (VGG-CIFAR-100), ResNet-20 (ResNet) (He et al., 2016) wihtout batch-normalization, a 6-layer CNN (6CNN), SimpleCNN used in Jastrzebski et al. (2021) (SimpleCNN) two 3-layer fully-connected networks (3FCN-CIFAR and 3FCN-MNIST), and WRN-28-10 for CIFAR-10/CIFAR-100 with the number of model parameters, $m = 9750922, 9797092, 268346, 511926, 361706, 656810, 199210, 36479194, 36536884$, respectively.

We use a modified version of the implementation of VGG-11 from `https://github.com/chengyangfu/pytorch-vgg-cifar10/blob/master/vgg.py` without the dropout layers and ResNet-20 from `https://github.com/locuslab/edge-of-stability/blob/github/src/resnet_cifar.py`. We change the last linear layer for the CIFAR-100 dataset. The 6CNN model can be expressed in the Pytorch code as follows:

```
nn.Sequential(
    nn.Conv2d(3, 32, 3, stride=1, padding=1, bias=False)
    nn.ReLU(),
    nn.Conv2d(32, 32, 4, stride=2, padding=1, bias=False)
    nn.ReLU(),
    nn.Conv2d(32, 64, 3, stride=1, padding=1, bias=False)
    nn.ReLU(),
    nn.Conv2d(64, 64, 4, stride=2, padding=1, bias=False)
    nn.ReLU(),
    nn.Flatten(),
    nn.Linear(4096, 100, bias=True),
    nn.ReLU(),
    nn.Linear(100, 10, bias=True),
)
```

and the 3FCN architecture is as follows:

```
nn.Sequential(
    nn.Flatten(),
    nn.Linear(n, 200, bias=True),
    nn.ReLU(),
    nn.Linear(200, 200, bias=True),
    nn.ReLU(),
    nn.Linear(200, 10, bias=True),
)
```

where `n=784` for 3FCN-MNIST, and `n=3072` for 3FCN-CIFAR (the same one used in Cohen et al. (2021)).

## E.3  HYPERPARAMETERS

SGD (Robbins & Monro, 1951) with the learning rate $\eta$ can be expressed as follows:

$$\boldsymbol{\theta}^{(t+1)} = \boldsymbol{\theta}^{(t)} - \eta \boldsymbol{g}^{\mathcal{B}^{(t)}}(\boldsymbol{\theta}^{(t)}) \tag{34}$$

where $\boldsymbol{\theta}^{(t)}$ is the model parameter, $\mathcal{B}^{(t)} \subset \mathcal{D}$ is the training batch at $t$-th step, $\boldsymbol{g} = \nabla_{\boldsymbol{\theta}} l$, and $\boldsymbol{g}^{\mathcal{B}} = \langle \boldsymbol{g} \rangle_{\mathcal{B}}$. We mostly use the simplest form of (S)GD as described in Eq (34) without momentum,

weight decay, and learning rate decay. The only exception is when we train WRN-28-10, we used the momentum of 0.9, the weight decay of 0.0005, and the learning rate decay of 0.2 in [30%, 60%, 80%] of the whole training epochs.

### E.4    DETAILED SETTINGS FOR EACH FIGURE

For Figure 2, 9, and 10, we ran the experiments on the CIFAR-10 dataset using 6CNN and trained the model using GD with the learning rate $\eta = 0.04$. See Appendix H for the setting used in Figure 6. We also plot variants of the figures in the main paper for other settings (training steps, data, network architectures, and hyperparameters) in Appendix J-L.

### E.5    HESSIAN

When computing the top eigenvalue $\|H\|_\sigma$ and the corresponding eigenvector $r$ of the Hessian $H$, we use the tool developed in PyHessian (Yao et al. (2020), `https://github.com/amirgholami/PyHessian`, MIT License) based on power iteration method using a small subset (5-25%) of training dataset $\mathcal{D}$.

### E.6    POWER ITERATION ALGORITHM

Even though we have the secular function $v(\lambda)$ in Eq (9) and the algorithms for computing the eigenvectors (Bunch et al., 1978), we use the power iteration in Algorithm 1 to get the top eigenvalue $\lambda^{(1)}$ of the logit Hessian $M \in \mathbb{R}^{C \times C}$ since we can run the algorithm for a mini-batch in parallel. Then, we can compute the corresponding top eigenvector $q^{(1)}$ from Theorem 1 (b). To compute the second largest eigenvalue, we apply the power iteration to $M' \equiv M - \lambda^{(1)} q^{(1)} q^{(1)T}$ instead of $M$ after computing $\lambda^{(1)}$ and $q^{(1)}$.

---

**Algorithm 1** Power iteration

    **Input:** matrix $M$, maximum iteration $n_{\max}$, tolerance bound $\epsilon$
    **Output:** the spectral norm $\lambda^{(1)} = \|M\|_\sigma$ of the matrix $M$
    Initialize $u \in \mathbb{R}^C$ with a random vector.
    $i \leftarrow 0$
    **repeat**
        $v \leftarrow Mu/\|Mu\|$
        $u \leftarrow M^T v/\|M^T v\|$
        $i \leftarrow i + 1$
    **until** it converges within the tolerance bound $\epsilon$ or $i \geq n_{\max}$
    **return** $v^T M u$

---

We also compute the operator norm of the Jacobian $\|\langle J \rangle\|$ with the power iteration as in 2. It requires $(C + 1)$-times scalar function differentiations with respect to $\theta$ for each iteration.

---

**Algorithm 2** Power iteration for Jacobian

---

**Input:** logit function $z_{\boldsymbol{\theta}}$ (not matrix $\langle \boldsymbol{J} \rangle$), maximum iteration $n_{\max}$, tolerance bound $\epsilon$
**Output:** the operator norm $\|\langle \boldsymbol{J} \rangle\|$ of the Jacobian
Initialize $\boldsymbol{u} \in \mathbb{R}^C$ with a random vector.
$i \leftarrow 0$
**repeat**
    Compute $\langle \boldsymbol{J} \rangle \boldsymbol{u} = \nabla_{\boldsymbol{\theta}} \langle \boldsymbol{u}^T \boldsymbol{z} \rangle$ ($1\times$ scalar function differentiation)
    $\boldsymbol{v} \leftarrow \langle \boldsymbol{J} \rangle \boldsymbol{u} / \|\langle \boldsymbol{J} \rangle \boldsymbol{u}\|$
    Compute $\langle \boldsymbol{J} \rangle^T \boldsymbol{v} = (\boldsymbol{v}^T \langle \boldsymbol{J} \rangle)^T = [\boldsymbol{v}^T \nabla_{\boldsymbol{\theta}} \langle \boldsymbol{z}_1 \rangle, \cdots, \boldsymbol{v}^T \nabla_{\boldsymbol{\theta}} \langle \boldsymbol{z}_C \rangle]^T$ ($C\times$ scalar function differentiations)
    $\boldsymbol{u} \leftarrow \langle \boldsymbol{J} \rangle^T \boldsymbol{v} / \|\langle \boldsymbol{J} \rangle^T \boldsymbol{v}\|$
    $i \leftarrow i + 1$
**until** it converges within the tolerance bound $\epsilon$ or $i \geq n_{\max}$
**return** $\boldsymbol{v}^T (\langle \boldsymbol{J} \rangle \boldsymbol{u})$

---

## F   THE TENDENCY OF THE JACOBIAN NORM TO INCREASE

The weight norm $\|\boldsymbol{\theta}^{(t)}\|$ increases (Figure 12 (the third one)) in order to increase the logit norm $\|\boldsymbol{z}(\boldsymbol{\theta}^{(t)})\|$ (Figure 12 (the leftmost figure)) and to minimize the cross-entropy loss during training (Soudry et al., 2018). This also leads to the increase in the layerwise weight norms (Figure 14) and the Jacobian norm (Figure 6). While the Jacobian norm tends to increase, it stops increasing at a certain point, stays for a few steps, and increases again. We ran the experiments on the CIFAR-10 dataset and 6CNN using GD with learning rate $\eta = 0.04$.

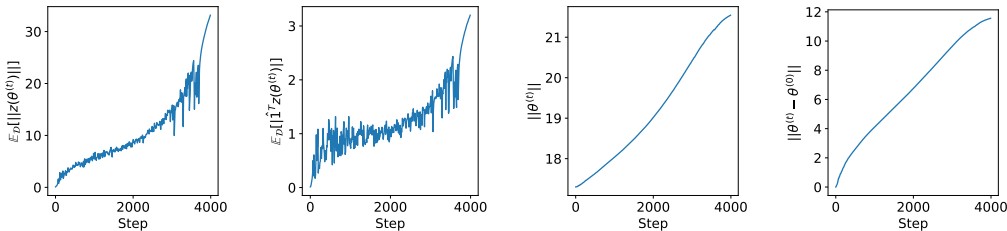

Figure 12: **(Left to Right): the logit norm** $\langle\|\boldsymbol{z}(\boldsymbol{\theta}^{(t)})\|\rangle$**, the absolute value of the logit sum** $\langle|\hat{1}^T\boldsymbol{z}(\boldsymbol{\theta}^{(t)})|\rangle$**, the weight norm** $\boldsymbol{\theta}^{(t)}$**, the distance from the initial weight** $\|\boldsymbol{\theta}^{(t)} - \boldsymbol{\theta}^{(0)}\|$ **during training (every 10 steps).** See together with Figure 6 (Left, solid red line) and Figure 5.

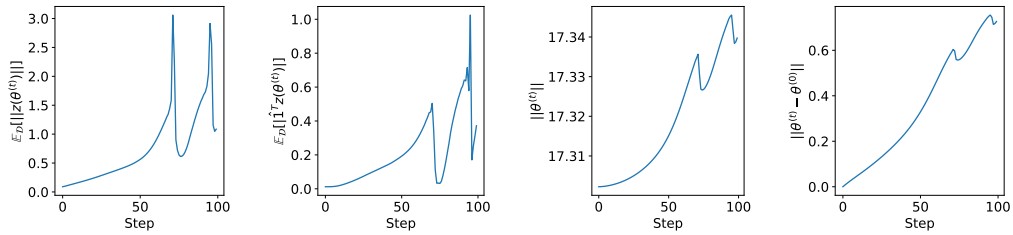

Figure 13: **(Left to Right): the logit norm** $\langle\|\boldsymbol{z}(\boldsymbol{\theta}^{(t)})\|\rangle$**, the absolute value of the logit sum** $\langle|\hat{1}^T\boldsymbol{z}(\boldsymbol{\theta}^{(t)})|\rangle$**, the weight norm** $\|\boldsymbol{\theta}^{(t)}\|$**, the distance from the initial weight** $\|\boldsymbol{\theta}^{(t)} - \boldsymbol{\theta}^{(0)}\|$ **in the early phase of training (every step).**

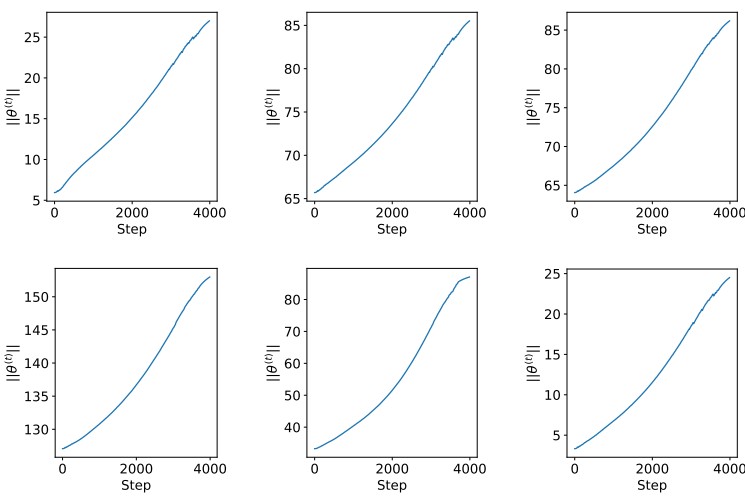

Figure 14: **The layerwise weight norms (6 layers from left to right and from top to bottom) of the 6CNN model in the early phase of training (every 10 step).**

# G $\quad \|\boldsymbol{H}\|_\sigma \propto \langle\lambda^{(1)}\rangle\|\langle\boldsymbol{J}\rangle\hat{\mathbf{1}}\|^2$

Surprisingly, we empirically observed that $\|\boldsymbol{H}\|_\sigma \propto \langle\lambda^{(1)}\rangle\|\langle\boldsymbol{J}\rangle\hat{\mathbf{1}}\|^2$ during training where $\hat{\mathbf{1}} = 1/\sqrt{C} \in \mathbb{R}^C$. Since it requires further theoretical grounding, we left it as future work. We observe this relation for a variety of learning rates (Figure 15), network architectures (Figure 16 and 17), batch sizes (Figure 18 and 19), and datasets (Figure 20 and 21). At least, $\|\boldsymbol{H}\|_\sigma$ and $\langle\lambda^{(1)}\rangle\|\langle\boldsymbol{J}\rangle\hat{\mathbf{1}}\|^2$, they increase and decrease together. We emphasize that Figure 21 shows the case when the sharpness did not reach the limit $2/\eta$. This is because the learning rate is relatively low and it is easy to train a model for the MNIST dataset, and thus $\langle\lambda^{(1)}\rangle$ decreases before the sharpness reaches the limit.

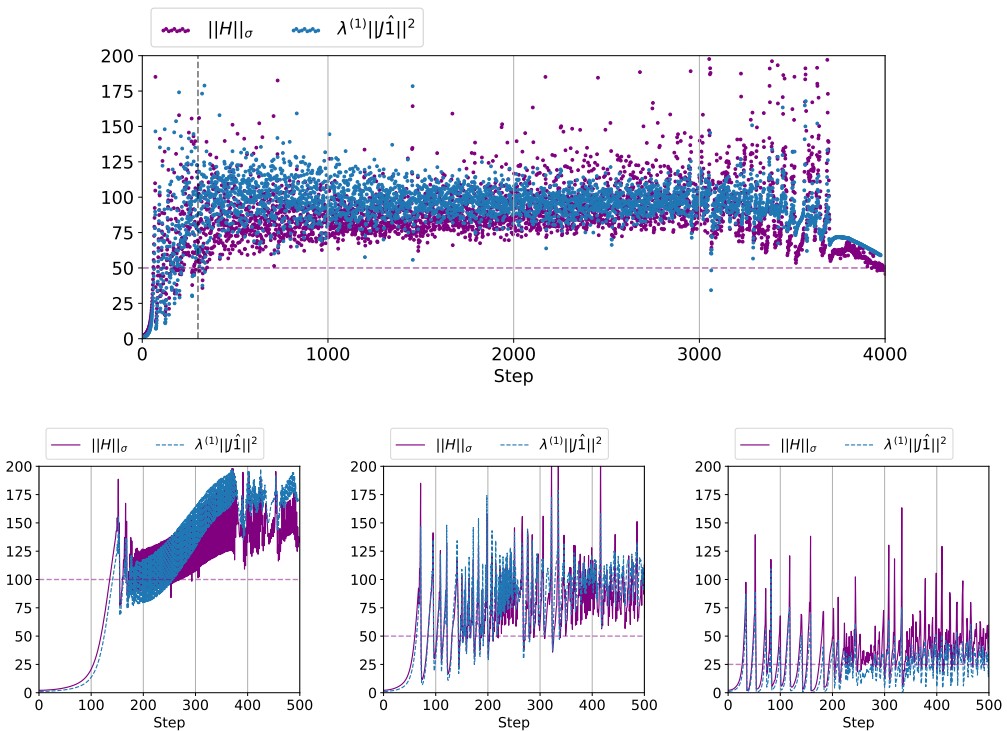

Figure 15: **The relation** $\|\boldsymbol{H}\|_\sigma \propto \langle\lambda^{(1)}\rangle\|\langle\boldsymbol{J}\rangle\hat{\mathbf{1}}\|^2$ **on the CIFAR-10 dataset and the 6CNN model trained using GD with different learning rates** $\eta = 0.02/0.04/0.08$ **(from left to right).** Bottom figures are plotted for the early phase of training. Top figure is plotted for $\eta = 0.02$. Curves are plotted for every step.

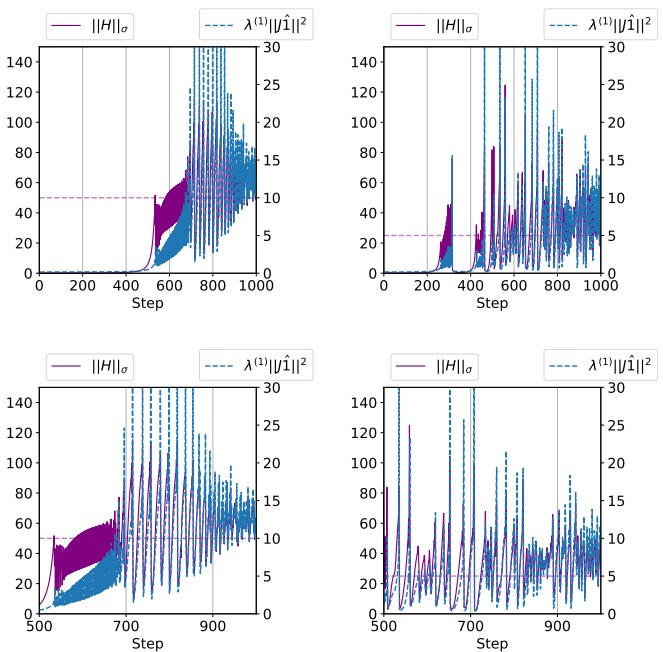

Figure 16: **The relation $\|H\|_\sigma \propto \langle\lambda^{(1)}\rangle\|\langle J\rangle\hat{1}\|^2$ on the CIFAR-10 dataset and the VGG model trained using GD with different learning rates $\eta = 0.04/0.08$ (left/right) in the early phase of training.** Curves are plotted for every step (Top: 0-1000 stpes and Bottom: 500-1000 stpes).

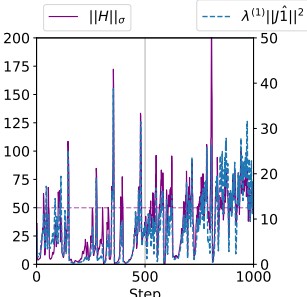

Figure 17: **The relation $\|H\|_\sigma \propto \langle\lambda^{(1)}\rangle\|\langle J\rangle\hat{1}\|^2$ on the CIFAR-10 dataset and the ResNet model trained using SGD with learning rate $\eta = 0.04$ and batch sizes $|\mathcal{B}^{(t)}| = 128$ during training (0-1000 steps).** Curves are plotted for every step.

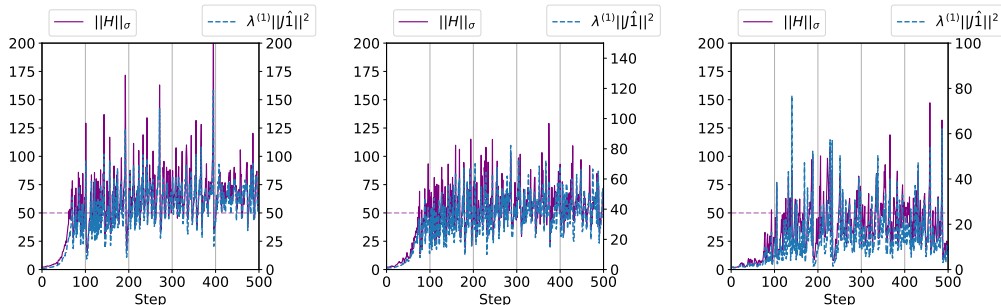

Figure 18: **The relation $\|\boldsymbol{H}\|_\sigma \propto \langle\lambda^{(1)}\rangle\|\langle\boldsymbol{J}\rangle\hat{\mathbf{1}}\|^2$ on the CIFAR-10 dataset and the 6CNN model trained using SGD with fixed learning rate $\eta = 0.04$ and different batch sizes $|\mathcal{B}^{(t)}| = 512/128/32$ (from left to right) in the initial phase (0-500 steps).** Note that the proportionality constant may change according to the batch size (the smaller the batch size, the larger the proportionality constant). Curves are plotted for every step.

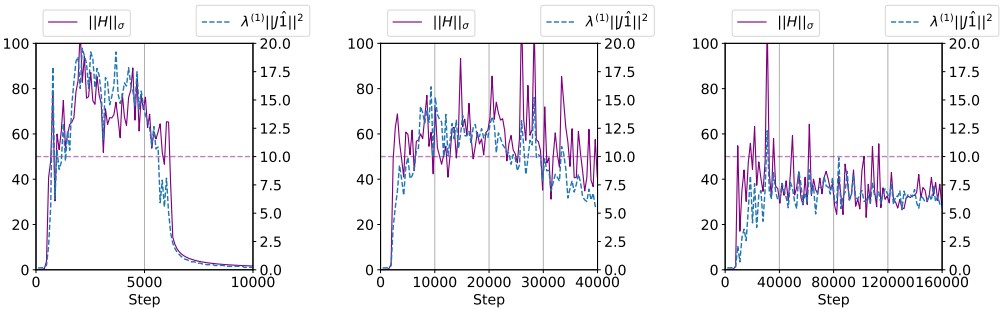

Figure 19: **The relation $\|\boldsymbol{H}\|_\sigma \propto \langle\lambda^{(1)}\rangle\|\langle\boldsymbol{J}\rangle\hat{\mathbf{1}}\|^2$ on the CIFAR-10 dataset and the VGG model trained using SGD with fixed learning rate $\eta = 0.04$ and different batch sizes $|\mathcal{B}^{(t)}| = 512/128/32$ (from left to right) during training (0-100 epochs).** Curves are plotted for every $n$ steps ($n = 97/388/1552$) where $97 = \lfloor 50000/512 \rfloor$.

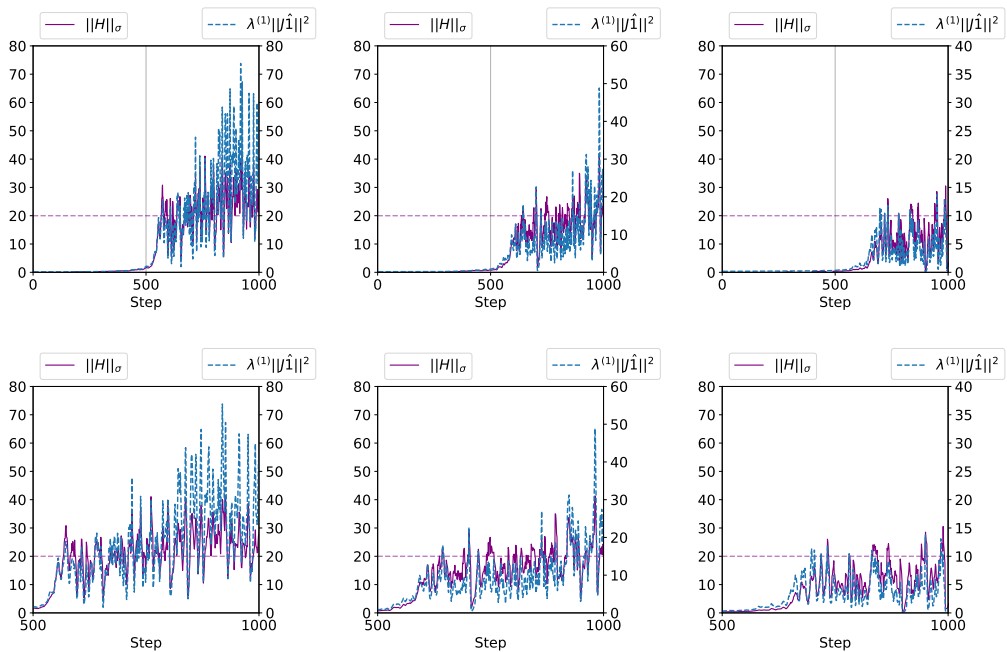

Figure 20: **The relation $\|H\|_\sigma \propto \langle\lambda^{(1)}\rangle\|\langle J\rangle\hat{1}\|^2$ on the CIFAR-100 dataset and the VGG model trained using SGD with fixed learning rate $\eta = 0.1$ and different batch sizes $|\mathcal{B}^{(t)}| = 128/64/32$ (from left to right) in the early phase of training.** Note that the proportionality constant may change according to the batch size (the smaller the batch size, the larger the proportionality constant). Curves are plotted for every step (Top: 0-1000 stpes and Bottom: 500-1000 stpes).

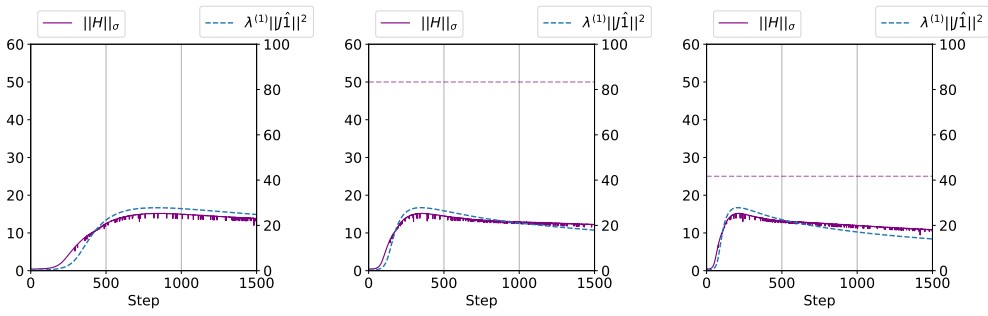

Figure 21: **The relation $\|H\|_\sigma \propto \langle\lambda^{(1)}\rangle\|\langle J\rangle\hat{1}\|^2$ on the MNIST dataset and the 3FCN-MNIST model trained using GD with different learning rates $\eta = 0.02/0.04/0.08$ (from left to right) in the early phase of training.** Note that the sharpness did not reach the limit $2/\eta$ (the dashed horizontal line). Curves are plotted for every step.

## H  IMPLICIT REGULARIZATION ON $\|\langle \boldsymbol{J} \rangle \hat{\boldsymbol{1}}\|^2$

We ran the experiments for Figure 2 on the CIFAR-10 dataset. We used 6CNN and VGG for Figure 2 (Left) and Figure 2 (Right), respectively.

We further investigate the relation $\|\boldsymbol{H}\|_\sigma \propto \langle \lambda^{(1)} \rangle \|\langle \boldsymbol{J} \rangle \hat{\boldsymbol{1}}\|^2$ in the previous section and its effect on $\|\langle \boldsymbol{J} \rangle \hat{\boldsymbol{1}}\|^2$. For the early phase of training, it is hard to see the initial rapid growth of the sharpness in this smoothed curves and when exactly the regularization begins to activate. We refer the readers to the previous section, Appendix G, for the fine-grained analysis of the early phase of training. We provide plots with different settings. Again, we observe that training with a larger learning rate and a smaller batch size limits $\|\langle \boldsymbol{J} \rangle \hat{\boldsymbol{1}}\|^2$ with a smaller value (dotted red lines) in the Active Regularization Period. Curves are smoothed for visual clarity.

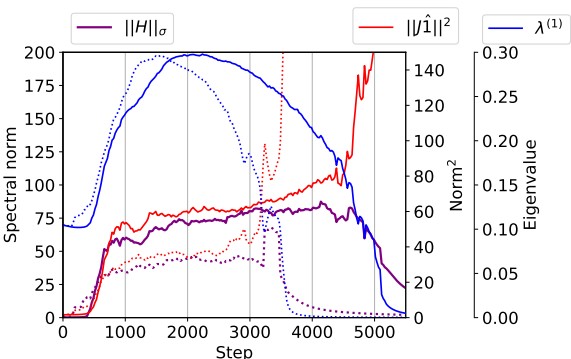

Figure 22: **The evolution of $\|\boldsymbol{H}\|_\sigma$, $\|\langle \boldsymbol{J} \rangle \hat{\boldsymbol{1}}\|^2$, and $\langle \lambda^{(1)} \rangle$ on the CIFAR-10 dataset and the VGG model trained using GD with the learning rates $\eta = 0.04/0.08$ (solid/dotted lines).** See the Figure 6 caption together.

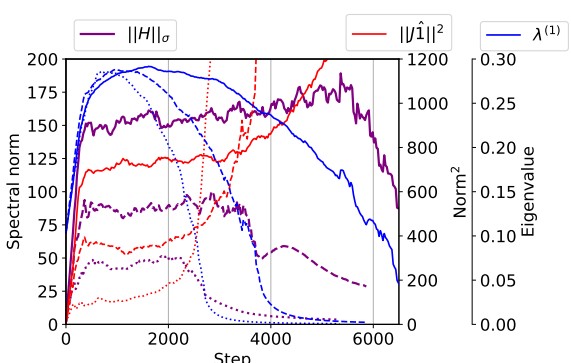

Figure 23: **The evolution of $\|\boldsymbol{H}\|_\sigma$, $\|\langle \boldsymbol{J} \rangle \hat{\boldsymbol{1}}\|^2$, and $\langle \lambda^{(1)} \rangle$ on the CIFAR-10 dataset and the 6CNN model trained using GD with the learning rates $\eta = 0.02/0.04/0.08$ (solid/dashed/dotted lines).** See the Figure 6 caption together.

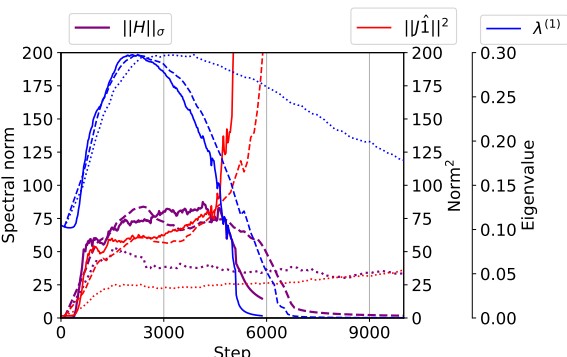

Figure 24: **The evolution of $\|H\|_\sigma$, $\|\langle J \rangle \hat{1}\|^2$, and $\langle \lambda^{(1)} \rangle$ on the CIFAR-10 dataset and the VGG model trained using SGD with the fixed learning rate $\eta = 0.04$ and the batch sizes $|\mathcal{B}^{(t)}| = 50000(\textbf{GD})/512/32$ (solid/dashed/dotted lines).** Training with a batch size 512 shows similar evolutions to the GD training. See the Figure 6 caption together.

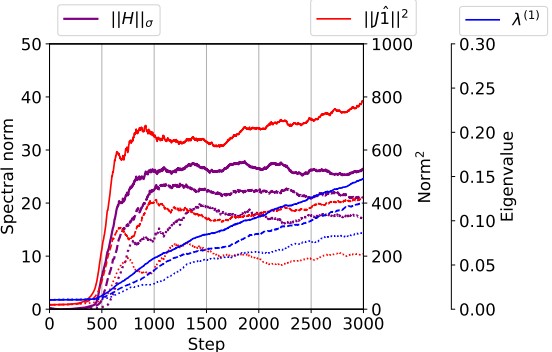

Figure 25: **The evolution of $\|H\|_\sigma$, $\|\langle J \rangle \hat{1}\|^2$, and $\langle \lambda^{(1)} \rangle$ on the CIFAR-100 dataset and the VGG model trained using SGD with the fixed learning rate $\eta = 0.1$ and the batch sizes $|\mathcal{B}^{(t)}| = 128/64/32$ (solid/dashed/dotted lines).** See the Figure 6 caption together.

## I FIGURE 4 (VISUALIZATION OF THE OPTIMIZATION TRAJECTORY)

We use UMAP (McInnes et al., 2018) to visualize the optimization trajectory $\{\boldsymbol{\theta}^{(t)}\}_{t\in[T]} \subset \Theta \subset \mathbb{R}^m$ in a 2D space. As GD enters into the Edge of Stability (Cohen et al., 2021), it oscillates in a direction nearly orthogonal to its global descent direction (Xing et al., 2018). Note that GD may not enter the Edge of Stability as shown in Figure 26 (Bottom).

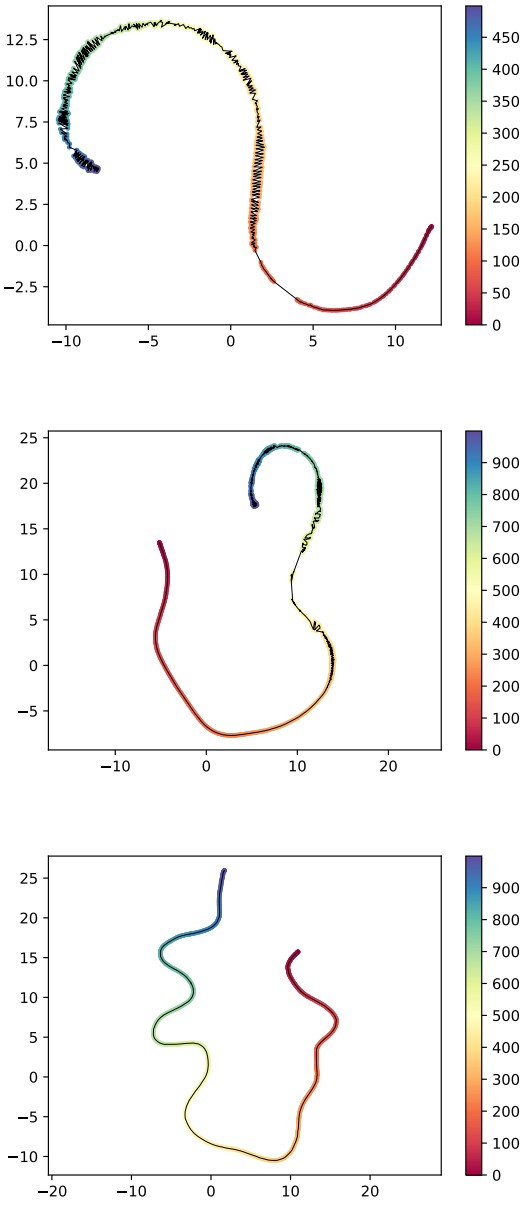

Figure 26: **Visualization of the optimization trajectory using UMAP.** (Top) UMAP on the CIFAR-10 dataset trained with 6CNN for the first 500 steps, (Middle) on the CIFAR-10 dataset trained with VGG for the first 1000 steps, and (Bottom) on the MNIST dataset trained with 3FCN-MNIST for the first 1000 steps (from red to blue).

## J  FIGURE 8 ($\boldsymbol{Q}\boldsymbol{\Lambda}^{1/2}$)

Figure 27 shows the matrix $\boldsymbol{Q}\boldsymbol{\Lambda}^{1/2}$ (Figure 8) for different training steps. Figure 8 and Figure 27 (Top Right) are plotted at the equivalent step ($t = 1000$). Figure 27 demonstrates that $\lambda^{(1)}$ becomes more dominant than the others as training progresses. The argument that there are two salient elements in each $\boldsymbol{q}^{(i)}$ in its ($i$)- and ($i+1$)-th elements is empirically shown to be valid throughout the training.

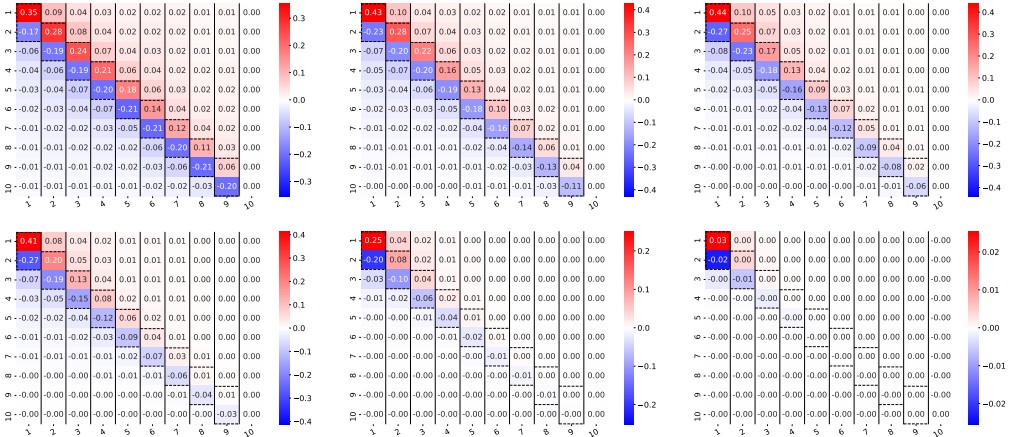

Figure 27: **Evolution of $\boldsymbol{Q}\boldsymbol{\Lambda}^{1/2}$ for some training steps during the training.** They are visualized for 100/500/1000/2000/4000/6000 steps from left to right and from top to bottom.

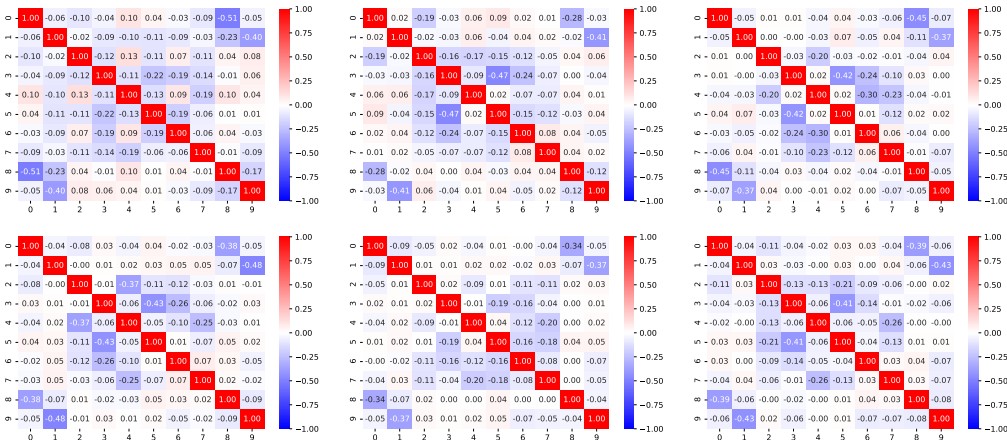

Figure 28: **Cosine similarities of $\{\overline{m}^i\}_{i=1}^C$ during training.** They are visualized for step=500/1000/1500/2000/3000/4000 from left to right and from top to bottom. See the Figure 9 caption.

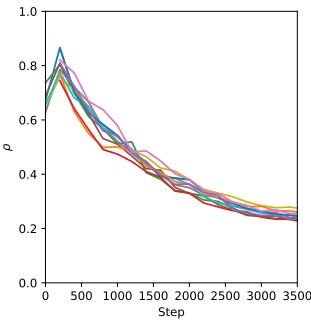

Figure 29: **The Mean Resultant Lengths (MRL) $\rho$ for each $\mathcal{D}_i$.** Note that $\rho$ is not defined for first few steps because some $\mathcal{D}_i$ are empty.

## K   FIGURE 9 (CLUSTERS OF $m$ AROUND EACH $\overline{m}^i$)

Figure 30 shows Figure 9 (Left) for some different class pairs $(i, j)$ with negative cosine similarity, i.e., $\cos(\overline{m}^i, \overline{m}^j) < 0$. We use the model trained for $t = 1000$ steps. Figure 28 shows Figure 9 (Right) for different training steps. Figure 29 shows the evolution of the Mean Resultant Length (MRL) $\rho$ in Figure 9 (Left) during training.

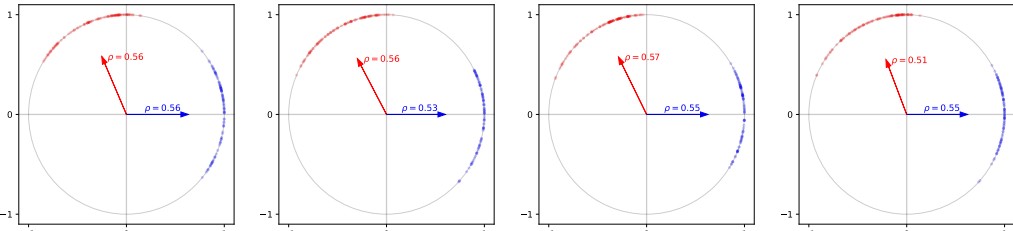

Figure 30: **Directional data of $m$ from $\mathcal{D}_i$ and $\mathcal{D}_j$.** They are visualized for $(i, j) =$ (airplane, ship)/(automobile, truck)/(dog, cat)/(deer, bird) from left to right. See the Figure 9 caption.

## L   FIGURE 10 (GRADIENT DESCENT HAPPENS MOSTLY IN THE TOP HESSIAN SUBSPACE)

Figure 31 shows similar results with different settings with VGG and learning rate $\eta = 0.08$.

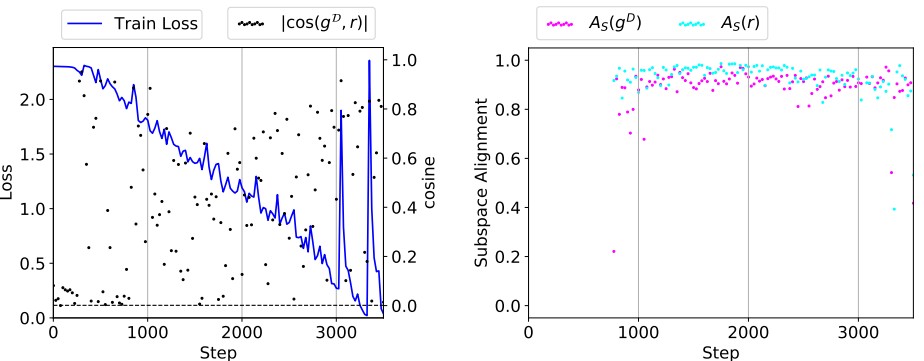

Figure 31: **(Left) Alignment between the two vectors $g^{\mathcal{D}}$ and $r$, and (Right) alignment of $g^{\mathcal{D}}, r$ in the subspace $\mathcal{S} = \text{span}(\{\overline{m}^i\}_{i=1}^C)$ using VGG with $\eta = 0.08$.** See the Figure 10 caption. They are plotted for every 25 steps. Note that $A_{\mathcal{S}}$ is not defined for first few steps (about 0-800 steps) because some $\mathcal{D}_i$ are empty.

# M ANALYSIS OF THE MSE LOSS

In the main text, we focus on the cross-entropy loss. Here, we briefly analyze the MSE loss, $l = \frac{1}{2}\|\boldsymbol{z} - \boldsymbol{e}^y\|^2$. Then, we have $\boldsymbol{M} = \nabla_{\boldsymbol{z}}^2 l = \boldsymbol{I}$, $\lambda^{(1)} = \|\boldsymbol{M}\|_\sigma = 1$ and $\boldsymbol{G} = \langle \boldsymbol{J}\boldsymbol{J}^T \rangle$. It leads to the same conclusion as in Theorem 2:

$$\|\boldsymbol{G}\|_\sigma = \|\langle \boldsymbol{J}\boldsymbol{J}^T \rangle\|_\sigma \leq \langle \|\boldsymbol{J}\boldsymbol{J}^T\|_\sigma \rangle = \langle \|\boldsymbol{J}\|^2 \rangle \tag{35}$$

We empirically observed that $\|\boldsymbol{H}\|_\sigma \propto \|\langle \boldsymbol{J} \rangle\|^2$ as shown in Figure 32.

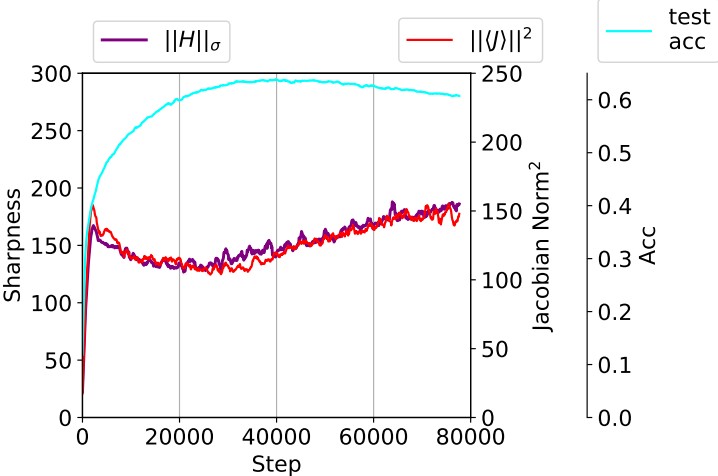

Figure 32: The sharpness $\|\boldsymbol{H}\|_\sigma$ and the Jacobian norm $\|\boldsymbol{J}\|^2$ during training with the MSE loss

# N    DETAILS OF EXPLICIT JACOBIAN REGULARIZATION (EJR)

We first propose a simple form of EJR with the regularized loss as follows:

$$\tilde{L}(\boldsymbol{\theta}) = L(\boldsymbol{\theta}) + \lambda_{reg}\|\langle \boldsymbol{J}\rangle\|_F^2/C \tag{36}$$

and update the model parameter as

$$\boldsymbol{\theta}^{(t+1)} = \boldsymbol{\theta}^{(t)} - \eta(\nabla_{\boldsymbol{\theta}}L(\boldsymbol{\theta}^{(t)}) + \lambda_{reg}\nabla_{\boldsymbol{\theta}}\|\langle \boldsymbol{J}\rangle\|_F^2/C) \tag{37}$$

However, this requires to build a computational graph for $\|\langle \boldsymbol{J}\rangle\|_F^2$ which is inefficient for a large network (e.g. WRN-28-10).

To this end, we propose two efficient variants of EJR. First, we propose to update the model parameter as follows:

$$\boldsymbol{\theta}^{(t+1)} = \boldsymbol{\theta}^{(t)} - \eta\nabla_{\boldsymbol{\theta}}L(\hat{\boldsymbol{\theta}}^{(t)}) \tag{38}$$

where $\hat{\boldsymbol{\theta}} = \boldsymbol{\theta} + \tilde{\rho}_{reg}\langle \boldsymbol{J}\rangle\boldsymbol{u}$ and $\tilde{\rho}_{reg} = \rho_{reg}/\|\langle \boldsymbol{J}\rangle\boldsymbol{u}\|$ as in Foret et al. (2021); Liu et al. (2019). Second, we propose another variant using $\hat{L}$ instead of $L$ in Eq (38) as follows:

$$\hat{L} = L + \mu_{reg}\boldsymbol{u}^T\langle \boldsymbol{z}\rangle \tag{39}$$

We can approximate $\hat{L}$ as follows:

$$\hat{L}(\hat{\boldsymbol{\theta}}) \approx \hat{L}(\boldsymbol{\theta}) + (\nabla_{\boldsymbol{\theta}}\hat{L}(\boldsymbol{\theta}))^T(\tilde{\rho}_{reg}\langle \boldsymbol{J}\rangle\boldsymbol{u}) \tag{40}$$

$$= \hat{L}(\boldsymbol{\theta}) + (\nabla_{\boldsymbol{\theta}}L(\boldsymbol{\theta}))^T(\tilde{\rho}_{reg}\langle \boldsymbol{J}\rangle\boldsymbol{u}) + (\nabla_{\boldsymbol{\theta}}\mu_{reg}\boldsymbol{u}^T\langle \boldsymbol{z}\rangle)^T(\tilde{\rho}_{reg}\langle \boldsymbol{J}\rangle\boldsymbol{u}) \tag{41}$$

$$= \hat{L}(\boldsymbol{\theta}) + (\nabla_{\boldsymbol{\theta}}L(\boldsymbol{\theta}))^T(\tilde{\rho}_{reg}\langle \boldsymbol{J}\rangle\boldsymbol{u}) + (\mu_{reg}\langle \boldsymbol{J}\rangle\boldsymbol{u})^T(\tilde{\rho}_{reg}\langle \boldsymbol{J}\rangle\boldsymbol{u}) \tag{42}$$

$$= \hat{L}(\boldsymbol{\theta}) + (\nabla_{\boldsymbol{\theta}}L(\boldsymbol{\theta}))^T(\tilde{\rho}_{reg}\langle \boldsymbol{J}\rangle\boldsymbol{u}) + \mu_{reg}\rho_{reg}\|\langle \boldsymbol{J}\rangle\boldsymbol{u}\| \tag{43}$$

$$\approx \hat{L}(\boldsymbol{\theta}) + (\nabla_{\boldsymbol{\theta}}L(\boldsymbol{\theta}))^T(\tilde{\rho}_{reg}\langle \boldsymbol{J}\rangle\boldsymbol{u}) + \mu_{reg}\rho_{reg}\|\langle \boldsymbol{J}\rangle\|_F/\sqrt{C} \tag{44}$$

We used the first-order Taylor expansion of $\hat{L}(\boldsymbol{\theta} + \hat{\rho}_{reg}\langle \boldsymbol{J}\rangle\boldsymbol{u})$ in Eq (40). We expect additional effect of minimizing $\mu_{reg}\boldsymbol{u}^T\boldsymbol{z} + \mu_{reg}\rho_{reg}\|\langle \boldsymbol{J}\rangle\|_F/\sqrt{C} \approx \mu_{reg}\rho_{reg}\|\langle \boldsymbol{J}\rangle\|_F/\sqrt{C}$ compared to the first variant.

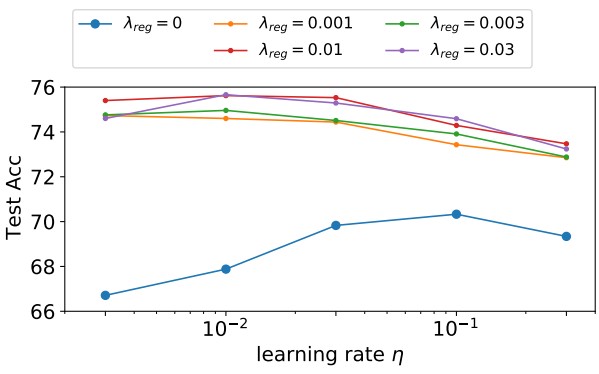

Figure 33: **[Explicit Jacobian Regularization] The explicit Jacobian regularization enhances the test accuracy.** We plot the test accuracy for different learning rates $\eta$ and regularization coefficients $\lambda_{reg}$. The models are trained with batch size of $|\mathcal{B}| = 128$ on CIFAR-10.

Table 2: **Effectiveness of EJR**. We report improvement (Δ**Acc.**) and Error Reduction Rate (**ERR**) on CIFAR-10 when trained with EJR (**+EJR**), compared to the standard training (**Baseline**).

| Dataset | Network Architecture | Batch Size | lr | Reg. param. | Test Accuracy | | ΔAcc. (%p) | ERR (%) |
|---|---|---|---|---|---|---|---|---|
| | | | | | Baseline | +EJR | | |
| CIFAR-10 | SimpleCNN | 128 | 0.003 | $\lambda_{reg}=0.01$ | 66.71 | **75.40** | +8.69 | 26.10 |
| | | | 0.01 | $\lambda_{reg}=0.03$ | 67.88 | **75.66** | +7.78 | 24.22 |
| | | | 0.03 | $\lambda_{reg}=0.01$ | 69.83 | **75.53** | +5.70 | 18.89 |
| | | | 0.1 | $\lambda_{reg}=0.03$ | 70.33 | **74.59** | +4.26 | 14.36 |
| | | | 0.3 | $\lambda_{reg}=0.01$ | 69.34 | **73.47** | +4.13 | 13.47 |
| | | 50000 (full-batch) | 0.01 | $\lambda_{reg}=0.001$ | 66.81 | **74.43** | +7.62 | 22.96 |
| | | | 0.03 | | 67.72 | **74.31** | +6.59 | 20.42 |
| | | | 0.1 | | 67.53 | **73.69** | +6.16 | 18.97 |
| | | | 0.3 | | 61.08 | **72.15** | +11.07 | 28.44 |
| | WRN-28-10 (200 epochs) | 128 | 0.1 | $\rho_{reg}=0.5$ | $95.93_{\pm0.15}$ | **96.44** | +0.51 | 12.72 |
| | | | | $\rho_{reg}=1$ | | **96.57** | +0.64 | 15.72 |
| | | | | $\rho_{reg}=2$ | | **96.62** | +0.69 | 16.95 |
| | | | | $\rho_{reg}=3$ | | **96.30** | +0.37 | 9.09 |
| | WRN-28-10 (400 epochs) | 128 | 0.1 | $\rho_{reg}=0.5$ | $96.10_{\pm0.05}$ | **96.65** | +0.55 | 14.10 |
| | | | | $\rho_{reg}=1$ | | **96.78** | +0.68 | 17.44 |
| | | | | $\rho_{reg}=2$ | | **97.07** | +0.97 | 24.87 |
| | | | | $\rho_{reg}=3$ | | **96.79** | +0.69 | 17.69 |
| CIFAR-100 | WRN-28-10 (200 epochs) | 128 | 0.1 | $\rho_{reg}=0.5$ | $80.29_{\pm0.25}$ | **80.42** | +0.13 | 0.66 |
| | | | | $\rho_{reg}=1$ | | **81.11** | +0.82 | 4.16 |
| | | | | $\rho_{reg}=2$ | | **81.50** | +1.21 | 6.14 |
| | | | | $\rho_{reg}=3$ | | **82.51** | +2.22 | 11.26 |
| | | | | $\rho_{reg}=4$ | | **82.65** | +2.36 | 11.97 |
| | | | | $\rho_{reg}=5$ | | **82.31** | +2.02 | 10.25 |
| | | | | $\rho_{reg}=6$ | | **82.03** | +1.74 | 8.83 |
| | WRN-28-10 (400 epochs) | 128 | 0.1 | $\rho_{reg}=1$ | $80.69_{\pm0.21}$ | **82.55** | +1.86 | 9.63 |
| | | | | $\rho_{reg}=2$ | | **82.51** | +1.82 | 9.42 |
| | | | | $\rho_{reg}=3$ | | **82.84** | +2.15 | 11.13 |
| | | | | $\rho_{reg}=4$ | | **83.35** | +2.66 | 13.78 |
| | | | | $\rho_{reg}=5$ | | **83.73** | +3.04 | 15.74 |
| | | | | $\rho_{reg}=6$ | | **83.16** | +2.47 | 12.79 |
| | | | | $\rho_{reg}=7$ | | **83.12** | +2.43 | 12.58 |
| | | | | $\rho_{reg}=8$ | | **81.16** | +0.47 | 2.43 |

Table 3: **Effectiveness of EJR.v2**. We report improvement (Δ**Acc.**) and Error Reduction Rate (**ERR**) on CIFAR-10 when trained with the second variant of EJR (**+EJR.v2**), compared to the standard training (**Baseline**).

| Dataset | Network Architecture | Batch Size | lr | Reg. param. | Test Accuracy | | ΔAcc. (%p) | ERR (%) |
|---|---|---|---|---|---|---|---|---|
| | | | | | Baseline | +EJR.v2 | | |
| CIFAR-10 | WRN-28-10 (400 epochs) | 128 | 0.1 | $\rho_{reg}=2, \mu_{reg}=0.001$ | $96.10_{\pm0.05}$ | **97.28** | +1.18 | 30.26 |
| | | | | $\rho_{reg}=2, \mu_{reg}=0.003$ | | **97.32** | +1.23 | 31.28 |
| | | | | $\rho_{reg}=2, \mu_{reg}=0.01$ | | **97.33** | +1.23 | 31.54 |
| | | | | $\rho_{reg}=2, \mu_{reg}=0.03$ | | **97.38** | +1.28 | 32.82 |
| | | | | $\rho_{reg}=2, \mu_{reg}=0.1$ | | **97.17** | +1.07 | 27.44 |
| | | | | $\rho_{reg}=1, \mu_{reg}=0.01$ | | **97.07** | +0.97 | 24.87 |
| | | | | $\rho_{reg}=1, \mu_{reg}=0.02$ | | **97.34** | +1.24 | 31.79 |
| | | | | $\rho_{reg}=1, \mu_{reg}=0.06$ | | **97.33** | +1.23 | 31.54 |
| | | | | $\rho_{reg}=1, \mu_{reg}=0.1$ | | **97.26** | +1.16 | 29.74 |

