# OpenReview forum: "Implicit Jacobian regularization weighted with impurity of probability output"
_ICLR.cc/2022/Conference — ICLR 2022 Submitted_

### Official Review · Reviewer_utPV · 2021-10-31

**Correctness:** 4
**Technical Novelty And Significance:** 2
**Empirical Novelty And Significance:** 2
**Recommendation:** 5
**Confidence:** 5

**Main Review:**

The paper is overall well written with clear mathematics.

Cons:  Theorem 1 is enlightening.

Weakness:  Can the author provide some analytical examples. E.g., how do the neural network structures and activation functions take effect in the logit Hessian matrix?  A  two-layer neural network example is needed.

In addition, there is essential literature on related information matrices, which are missed.

Li, Zhao, Wasserstein information matrix, arXiv:1910.11248

**Summary Of The Paper:**

The authors study the largest eigenvalue and eigenvector of the Hessian of the loss function. The authors approximate the Hessian matrix by a low-dimensional matrix, which is a rank-one modification of a diagonal matrix. The eigendecomposition helps to explain how the sharpness influences the gradient descent method and the generalization properties. They show that GD has implicit regularization effects on the Jacobian norm weighted with the impurity of the probability output, which is also related to the Fisher information matrix.

**Summary Of The Review:**

The paper proposes to study an approximate Hessian operator of the loss function. This is helpful in understanding the fundamental non-convexity questions in learning.

A more analytical example is needed to improve the current paper. The example should address how the simplified Hessian matrix depends on the neural network functions and architectures.

---

> ### Author Response · Authors · 2021-11-17
> **Answer**
>
> We summarize what we have updated in the 1st revised version as follows:
> - Notations
>    - To avoid confusion, we use the notation $\langle a\rangle$ for the expected value $\mathbb{E}_\mathcal{D}[a]$.
>    - We use the notation $||A||$ for the operator norm which is equivalent to the spectral norm $||A||_\sigma$ for square matrices.
> - Theorem
>    - The previous discussions on $||J1||$ (or $||\langle J\rangle 1||$) lack theoretical grounding.
>    - Thus, we provide another approach with the operator norm $||\langle J\rangle||^2$ ("the Jacobian norm") with a new Theorem (Thm 2).
>    - We also provide how to estimate the operator norm of the Jacobian in Appendix E (power iteration).
> - EJR
>    - We propose two efficient variants of EJR to train a larger model (WRN-28-10) as detailed in Appendix N.
>    - For a larger model, theses EJR show noticeable performance improvements (Tab 1 and Fig 7).
> - IJR
>    - We analyze the operator norm $||\langle J\rangle||$ and the implicit bias in SGD.
>    - Unlike $||\langle J\rangle 1||$, the operator norm $||\langle J\rangle||$ does not show a step-like structure with plateau, but it still shows three phases of IJR.
>    - We provide a plot (Fig 5) of IJR with loss and accuracy.
> - MSE
>    - In Appendix M, we provide a discussion on the sharpness and the Jacobian norm when trained with the MSE loss instead of the cross-entropy loss.
>
> C1. Can the author provide some analytical examples. E.g., how do the neural network structures and activation functions take effect in the logit Hessian matrix? A two-layer neural network example is needed.
> - A1. It is unnecessary. The logit Hessian is defined as $M=\nabla^2_z p$ and we proved that $M=\text{diag}(p)-pp^T$. Thus, it is not affected by the neural network structure, but only by the output probability $p$.

---

> > ### Author Response · Authors · 2021-11-22
> > **Answer 2**
> >
> > C2. In addition, there is essential literature on related information matrices, which are missed. Li, Zhao, Wasserstein information matrix, arXiv:1910.11248
> > - A2.
> >    - We understand the suggested paper is an essential literature, but we are not sure how it is related with ours. Could you please elaborate more on how the suggested paper is related with our paper? We hope to get some useful insights from the suggested paper.
> >    - We cannot list all the papers about FIM because there are too many of those and our focus is not on FIM itself, but to figure out what factors are implicitly influenced by SGD. We reviewed [1] because it empirically showed that SGD implicitly penalizes the trace of the FIM.
> >
> > [1] Jastrzebski et al., 2021, Catastrophic fisher explosion: Early phase fisher matrix impacts generalization

---

### Official Review · Reviewer_4LVi · 2021-11-01

**Correctness:** 2
**Technical Novelty And Significance:** 3
**Empirical Novelty And Significance:** 3
**Recommendation:** 5
**Confidence:** 3

**Main Review:**

Strengths

1. This study links three concepts in optimization and generalization (flat minima, oscillation, gradient regularization). It is interesting to see how the diversity (i.e. entropy) of the softmax outputs is connected to the sharpness of the loss landscape.
2. The eigenvalues of the approximated Hessian (logit Hessian) are theoretically analyzed (Theorem 1).
3. There are experiments that are intended to support the theoretical analysis.


Weaknesses

1. This paper often lacks clear descriptions of technical details that are necessary to be contained. I read this paper three times from the beginning, but I'm still puzzled. My questions are:
- What is the experimental setting of Figure 3? Appendix C4 says CNN is trained on CIFAR10, but the caption says MNIST and Resnet are also used.
- What is the model architecture of Figure 4?
- In Eq. (11), the expectation \hat{E}_D is separately applied to \lambda and J. Why you can do that? I concern this operation yields a significant approximation error. Indeed, in Figure 5, at the point around 3500 steps ||J1|| reaches 1000 and \lambda^{(1)} takes around 0.1, which means ||H|| should be around 100 according to (11). However, the actual value is 50. This gap looks large.
- There is a sentence "This explains why the behavior of the sharpness in the early phase of training seems to impact the final generalization." in Section 5.2. Could you elaborate on this? Why the sharpness in the early phase influences the final performance?
- The same paragraph says "λ acts as a regularization coefficient". Please clarify this. Do you mean the objective is approximated as `original_loss + λ ||H||_σ`? If so, please derive this mathematically.
2. Jacobian regularization is not significantly evaluated. This study suggests using the Frobenius norm of the gradient as a regularizer. However, its effect is not comprehensively examined in an empirical manner. How is it better than popular regularizers (e.g. weight decay, l1, gradient penalty, Jacobian regularizers described in [*])? How is the effect consistent for different architectures and datasets?
3. The motivation is not solid. As is described in Section 2, there are several studies working on the decomposition of the Hessian (e.g. Papyan (2019)). Why the decomposition H ~= JMJ should be studied in this paper?


[*] https://www.ejournals.eu/pliki/art/13926/

**Summary Of The Paper:**

In this paper, the relationship between the Jacobian (the gradient of the final activation w.r.t parameters) and the Hessian is analyzed for the softmax cross-entropy loss. As a key tool, the approximation of the Hessian with the probability vector (the softmax output) is used, which suggests connections with several optimization (and generalization) concepts such as sharpness v.s. flatness of the loss landscape and the oscillation on the optimization path at GD/SGD.

**Summary Of The Review:**

This paper provides a new view that connects three notions in optimization and generalization (flat minima, oscillation, gradient regularization) with theoretical and empirical supports. However, the current manuscript lacks clear descriptions; without these, I cannot evaluate the correctness. Furthermore, the proposed method (Jacobian regularization) is not significantly evaluated. Also, the motivation of using the proposed decomposition is not grounded.

---

> ### Author Response · Authors · 2021-11-17
> **Thank you for your insightful and valuable comments.**
>
> Thank you for your insightful and valuable comments. We have tried our best to improve our paper by taking your comments into consideration. We have revised our paper and please kindly check it. We highlighted the changes (in red text) in the revised version.
>
> We summarize what we have updated in the 1st revised version as follows:
> - Notations
>    - To avoid confusion, we use the notation $\langle a\rangle$ for the expected value $\mathbb{E}_\mathcal{D}[a]$.
>    - We use the notation $||A||$ for the operator norm which is equivalent to the spectral norm $||A||_\sigma$ for square matrices.
> - Theorem
>    - The previous discussions on $||J1||$ (or $||\langle J\rangle 1||$) lack theoretical grounding.
>    - Thus, we provide another approach with the operator norm $||\langle J\rangle||^2$ ("the Jacobian norm") with a new Theorem (Thm 2).
>    - We also provide how to estimate the operator norm of the Jacobian in Appendix E (power iteration).
> - EJR
>    - We propose two efficient variants of EJR to train a larger model (WRN-28-10) as detailed in Appendix N.
>    - For a larger model, theses EJR show noticeable performance improvements (Tab 1 and Fig 7).
> - IJR
>    - We analyze the operator norm $||\langle J\rangle||$ and the implicit bias in SGD.
>    - Unlike $||\langle J\rangle 1||$, the operator norm $||\langle J\rangle||$ does not show a step-like structure with plateau, but it still shows three phases of IJR.
>    - We provide a plot (Fig 5) of IJR with loss and accuracy.
> - MSE
>    - In Appendix M, we provide a discussion on the sharpness and the Jacobian norm when trained with the MSE loss instead of the cross-entropy loss.
>
> We will provide detailed reply in separate comments.

---

> ### Author Response · Authors · 2021-11-18
> **Answer 1/2**
>
> Before we start, we would like to summarize the paper.
>
> - Deep neural networks are trained using SGD with discrete step sizes.
> - This discrete step size and the catapult mechanism limit the sharpness below a threshold.
> - As shown in the Hessian decomposition, this implicit effect on the sharpness also affects and regularizes the Jacobian norm.
> - We argue that this effect on the Jacobian norm is a main reason why SGD leads to minima that generalize well.
> - To support this claim, we explicitly regularized the Jacobian norm and observed significant performance improvement.
>
> ------
> We use new reference numbers for Figures, Tables, Sections, etc. used in the 1st revised version. Please kindly check the revised manuscript.
>
> C1-2. What is the model architecture of Figure 4?
> - A1-2. As detailed in Appendix, it is 6CNN. We also provide equivalent plots for VGG on CIFAR-10 and 3FCN on MNIST.
>
> C1-3-2.  Indeed, in Figure 5, at the point around 3500 steps ||J1|| reaches 1000 and \lambda^{(1)} takes around 0.1, which means ||H|| should be around 100 according to (11). However, the actual value is 50. This gap looks large.
> - A1-3-2.
>    - We updated the manuscript with a new relation because the previous relation lacks theoretical groundings, please kindly check the revised version.
> However, the gap is actually not that large.
> See Figure 15 for exact value of $\langle \lambda^{(1)}\rangle ||\langle J\rangle 1||^2$ in each step.
> While it is true that $\langle \lambda^{(1)}\rangle ||\langle J\rangle 1||^2$ overestimates $||H||_\sigma$ in the later phase of training, the gap is actually not that large. We also note that $||H||_\sigma\approx ||G||_\sigma$ sometimes fails to hold in the later phase, thus we focus on the early phase of training.
>    - Also please note that $||H||_\sigma$ and $\langle \lambda^{(1)}\rangle ||\langle J\rangle 1||^2$ behave similarly **up to a constant factor**. For example, see Figure 16, 17, 18, 19, 20, 21. They have different left and right y-axes.
>
>
>
> C1-4. There is a sentence "This explains why the behavior of the sharpness in the early phase of training seems to impact the final generalization." in Section 5.2. Could you elaborate on this? Why the sharpness in the early phase influences the final performance?
> - A1-4.
>    -  In the early phase, the sharpness increases, reaches the limit, and is limited near below the limit.
> Thus the Jacobian norm is regularized due to the sharpness-Jacobian-impurity relation. With a large $\eta$, the Jacobian norm is actively regularized with the sharpness being a small value near below $2/\eta$.
> In the later phase, decrease in the impurity $\lambda^{(1)}$ leads to diminishing the regularization effect. Thus, the early phase is important.
>
> C1-5. The same paragraph says "λ acts as a regularization coefficient". Please clarify this. Do you mean the objective is approximated as original_loss + λ ||H||_σ? If so, please derive this mathematically.
> - A1-5.
>    - In the revised version, we said “$ \lambda^*||J||^2$ has the expected value of approximately $||H||\_\sigma$, and we briefly state this relation with $\lambda^*||J||^2\sim||H||\_\sigma$. Here, $\lambda^\ast$ in $\lambda^\ast ||J||^2$ acts as an adaptive regularization weight for the regularization on the Jacobian norm $||J||^2$” and “the regularization effect diminishes as $\lambda^{(1)}$ decreases (the upper limit $\frac{2}{\eta\lambda^\ast}$ is loosened) with increasing $p_{(1)}\geq0.5$ in the later phase (see Figure 2, 5 and 6) because $\lambda^\ast\leq\lambda^{(1)}$ acts as a regularization coefficient.”
>    - We meant that since the expected value ($||H||_\sigma$) is limited, $ \lambda^\ast||J||^2$ is also limited for each $x$.  Since $\lambda^\ast||J||^2\leq\lambda^{(1)}||J||^2$, decrease in $\lambda^\ast$ (decrease in $\lambda^{(1)}$) in the later phase allows room for $||J||^2$ to increase. We said “$\lambda^\ast\leq\lambda^{(1)}$ acts as a regularization coefficient” in this sense.
>    - One may understand it as “the objective is approximated as $L+ \lambda||H||\_\sigma$” in the sense that a gradient-based optimization  with a discrete step size minimizes the original loss subject to the constraint that $||H||\_\sigma\lessapprox 2/\eta$. We can obtain “$L+ \lambda||H||\_\sigma$” as a **Lagrangian** function.
>    - In Explicit Jacobian Regularization (EJR), we used a regularized loss, $L+\lambda_{reg}||\langle J\rangle ||^2_F/C$.
> There are some variants of EJR introduced in the paper.

---

> > ### Author Response · Authors · 2021-11-18
> > **Answer 2/2**
> >
> > C1-1. What is the experimental setting of Figure 3? Appendix C4 says CNN is trained on CIFAR10, but the caption says MNIST and Resnet are also used.
> > - A1. We updated the manuscript. We used a variety of lr (Fig 15), network architecture (Fig 16,17), batch sizes (Fig 18,19), and datasets (Fig 20,21) in Appendix G (revised version).
> >
> > C1-3-1. In Eq. (11), the expectation \hat{E}_D is separately applied to \lambda and J. Why you can do that? I concern this operation yields a significant approximation error.
> > - A1-3-1.
> >    - We can compute $\langle J\rangle u=\nabla_\theta (u^T \langle z \rangle)$ easily because it is a single scalar differentiation, but $\langle Ju \rangle$ requires $\mathcal{D}$-times differentiations for each $Ju=\nabla_\theta (u^T z)$. Please kindly see the power iteration algorithm in Appendix E for the Jacobian (operator) norm, which requires further computation.
> >    - We updated the manuscript. In Theorem 2, we proved that $||H||_\sigma\approx||G||_\sigma=||\langle JMJ^T\rangle||_\sigma=\langle \lambda^\ast||J||^2\rangle\leq\langle \lambda^{(1)}||J||^2\rangle$. Thus, because of the plateau of the sharpness, the Jacobian norm $||J||^2$ is regularized for each $x\in\mathcal{D}$.
> > Since it is computationally inefficient to track $||J||$ for every $x\in\mathcal{D}$, we instead investigate $||\langle J \rangle||^2$. We expect the (expected) Jacobian norm $||\langle J\rangle||^2$ to be regularized.
> >    - The expected $J$ operation may yield an approximation error, but it does not affect the relation much since $||\langle J\rangle||^2$ is also regularized when $||J||^2$ is regularized for each $x$.
> >
> > C2. Jacobian regularization is not significantly evaluated. This study suggests using the Frobenius norm of the gradient as a regularizer. However, its effect is not comprehensively examined in an empirical manner. How is it better than popular regularizers (e.g. weight decay, l1, gradient penalty, Jacobian regularizers described in [*])? How is the effect consistent for different architectures and datasets?
> > - A2. Table 1 and Figure 7 show some experimental results of the effectiveness of EJR.
> >    - Our main focus was not to achieve a SOTA model, but to support the claim that SGD implicitly regularizes the Jacobian norm and it is a main reason why SGD leads to minima that generalize well.
> > To support the claim, we showed the performance improvement when introducing EJR.
> > We did not compare it with other methods for these reasons, but we will do in the future.
> >    - The following is some previous reported SOTA results (**proposed/baseline** on **CIFAR-10 and CIFAR-100**, respectively) conducted on WRN-28-10 (a single method and a single model). Note that this is not a fair comparison as they may use different training hyperparameters, but it shows that EJR achieved a comparable performance to SOTA.
> >       - EJR:
> >          - $97.38/96.10$ and $83.98/80.87$ (will be updated in the later version)
> >       - Manifold Mixup [1]:
> >          - $97.45/96.01$ and $81.96/78.28$
> >       - SWA [2]:
> >          - $96.79/96.18$ and $82.15/80.82$
> >       - LPBN [3]:
> >          - $96.46/96.16$ and $82.87/81.35$
> >       - BA [4]:
> >          - $97.15/96.60$ and $83.45/79.85$
> >       - PuzzleMix [5]:
> >          - $84.05/78.86$ on CIFAR-100 (CIFAR-10 not available)
> >       - SAM [6]:
> >          - $97.6/96.5$ and $83.7/80.9$ (1800 epochs).
> >          - $97.3/96.5$ and $83.5/81.2$ (200 epochs).
> >
> > C3. The motivation is not solid. As is described in Section 2, there are several studies working on the decomposition of the Hessian (e.g. Papyan (2019)). Why the decomposition H ~= JMJ should be studied in this paper?
> > - A3
> >     - With a different decomposition, we analyze the Hessian from another point of view.
> >     - Especially, we investigated the eigendecomposition of the logit Hessian $M$, while Papyan (2019) used a different decomposition of $M$.
> >     - With the eigendecomposition, we can obtain Theorem 2 which explains why SGD leads to minima that generalize well and why the early phase of training is more important for generalization than the later phase [7].
> >
> > **Reference**
> >
> > [1] Verma et al., 2018, Manifold Mixup: Better Representations by Interpolating Hidden States (https://arxiv.org/abs/1806.05236)
> >
> > [2] Izmailov et al., 2018, Averaging Weights Leads to Wider Optima and Better Generalization (https://arxiv.org/abs/1803.05407)
> >
> > [3] Von Oswald et al., 2021, NEURAL NETWORKS WITH LATE-PHASE WEIGHTS (https://arxiv.org/pdf/2007.12927v3.pdf)
> >
> > [4] Hoffer et al., 2020, Augment Your Batch: Improving Generalization Through Instance Repetition (https://arxiv.org/abs/1901.09335)
> >
> > [5] Kim et al., 2020, Puzzle Mix: Exploiting Saliency and Local Statistics for Optimal Mixup (https://arxiv.org/pdf/2009.06962v2.pdf)
> >
> > [6] Foret et al., 2020, Sharpness-Aware Minimization for Efficiently Improving Generalization  (https://arxiv.org/abs/2010.01412)
> >
> > [7] Achille et al. 2017, Critical Learning Periods in Deep Neural Networks (https://arxiv.org/abs/1711.08856)

---

> > > ### Comment · Reviewer_4LVi · 2021-11-24
> > > **Thank you for your response**
> > >
> > > I read your responses and other review comments. By taking into account all of them, I decided to keep my original score.
> > > - My questions about experimental settings were resolved.
> > > - The technical part was not convincing to me. It seems the theoretical results and empirical results that should be distinguished are mixedly used to conclude a vague statement. For example, A1-3-1 noted "||H|| and λ||J1|| behave similarly up to a constant factor", but what does it mean exactly? Can you prove ||H|| = \Theta(λ||J1||)? Or just saying this by showing empirical results?
> > > - The motivation part was still unclear. Why should we consider thinking about the "different" decomposition here? Are the existing decompositions not sufficient? I checked Theorem 2 but the sentence "Theorem 2 which explains why SGD leads to minima that generalize well and why the early phase of training is more important for generalization than the later phase" sounds overstating.

---

> > > > ### Author Response · Authors · 2021-11-24
> > > > **Thank you for additional valuable comments**
> > > >
> > > > C1. The technical part was not convincing to me. It seems the theoretical results and empirical results that should be distinguished are mixedly used to conclude a vague statement. For example, A1-3-1 noted "||H|| and λ||J1|| behave similarly up to a constant factor", but what does it mean exactly? Can you prove ||H|| = \Theta(λ||J1||)? Or just saying this by showing empirical results?
> > > > - A1.
> > > >    - We are sorry for the confusion. We have no theoretical proof of the empirical result "$||H||\_\sigma$ and $\langle \lambda^{(1)}\rangle||\langle J\rangle 1||^2$ behave similarly up to a constant factor".
> > > >    - Thus, we replaced this argument with "$||H||\_\sigma\approx \langle \lambda^\ast  ||J||^2\rangle\leq \langle \lambda^{(1)} ||J||^2\rangle$" in Theorem 2.
> > > >
> > > > C2. The motivation part was still unclear. Why should we consider thinking about the "different" decomposition here? Are the existing decompositions not sufficient? I checked Theorem 2 but the sentence "Theorem 2 which explains why SGD leads to minima that generalize well and why the early phase of training is more important for generalization than the later phase" sounds overstating.
> > > > - A2.
> > > >    - The crucial difference of ours from the existing decompositions is the eigendecomposition of the logit Hessian $M$.
> > > >    - The top eigenvalue $\lambda^{(1)}$ of the logit Hessian becomes smaller in the later phase.
> > > >    - The plateau of the sharpness and the catapult mechanism explain how discrete dynamics of SGD with a finite step size ($\eta$) implicitly regularizes the sharpness $||H||_\sigma$ of the loss landscape. Roughly speaking, with a finite step size, SGD cannot explore high-sharpness area.
> > > >    - Thus, Theorem 2 explains that SGD implicitly regularizes the squared norm of the Jacobian, $||J||^2$, weighted with $\lambda^\ast\leq\lambda^{(1)}$.
> > > >    - As $\lambda^{(1)}$ becomes smaller in the later phase, the regularization effect diminishes. Therefore, the early phase of training is more important for generalization than the later phase.
> > > >    - We argue that the regularization effect on the Jacobian norm is a main reason why SGD leads to minima that generalize well. To support this claim, we explicitly regularized the Jacobian norm and observed significant performance improvement.
> > > >    - To summarize, Theorem 2 explains (1) why SGD leads to minima that generalize well
> > > >         - answer: SGD regularizes the Jacobian norm ("$||H||\_\sigma\approx \langle \lambda^\ast  ||J||^2\rangle$") and its effect on the generalization is supported by EJR
> > > >    - and (2) why the early phase of training is more important for generalization than the later phase.
> > > >         - answer: the regularization effect diminishes in the later phase of training with a small $\lambda^\ast\leq\lambda^{(1)}$ ("$||H||\_\sigma\approx \langle \lambda^\ast  ||J||^2\rangle\leq \langle \lambda^{(1)} ||J||^2\rangle$") which allows room for $||J||$
> > > >  to increase.

---

### Official Review · Reviewer_6UXZ · 2021-11-01

**Correctness:** 2
**Technical Novelty And Significance:** 1
**Empirical Novelty And Significance:** 2
**Recommendation:** 3
**Confidence:** 4

**Main Review:**

- Theorem 1: What is novel in the claim and proof compared to prior work? In particular, I am referring here to the contributions from linear algebra by Golub (1973), Golub & van Loan, and Bunch et al. (1978). For example, Eq. (9) is known as the secular equation of the eigen-decomposition problem for rank-one modification of a diagonal matrix. Results (a-c) are for sure quite well-known in that literature (this might be a misrepresentation of the theoretical contributions). So, the novel contribution here might only be (d), which should then be formulated as a separate proposition and Theorem 1 should be cited with Golub and others.

- Section 4.2: In the introduction, there has been a lot of stress on eigenvalues of the Hessian, which should be matrix H. This section, on the other hand, states that the evolution of top eigenvalues of matrix M are tracked. This is the matrix of an exponential family model that completely ignores all the layers of a neural network apart from the last one where the softmax operates. The relation to prior work by Cohen et al (2021) is also unclear -- what would be a novel finding compared to that work? In what sense, the current work extends the estimation procedure from that work?

- Figure 3 and opening of Section 5 is unclear and too informal for a conference publication. The appendix is optional and findings should be described clearly in the main part of the paper.

- Section 5.1: Impurity here refers to the top eigenvalue of M? This notion is not defined, there is just a bound of the top eigenvalue in terms of Gini impurity.

- Eq. (11): I did not follow the derivation for this?

- Section 5.2: Could you please clarify the implication that weight norm increase leads to increase in the Jacobian norm? The Jacobian norm should be about the geometry of the loss landscape which can be independent of the actual parameter values? Alternatively, if you start with all-zero parameters then this is to be expected.

- The reasoning on the Jacobian regularization is unclear. Would not Jacobian norm need to decrease for GD to be able to converge?

**Summary Of The Paper:**

The paper investigates the dynamics of leading eigenvalues in the so called logit-Hessian matrix and tries to correlate that with the convergence to a good local optima. The work also tracks the behavior of the Jacobian matrix for the pre-softmax outputs and introduces the notion of implicit Jacobian regularization.

**Summary Of The Review:**

The paper is poorly written and too informal in a number of places. There are many claims which are presented as facts but are in reality hypotheses. I have expected an analysis based on the Hessian of a neural network and it turns out that the work focuses only on the Hessian matrix of the exponential family model, completely ignoring the network parameters up to the sofmax operator. I also fail to grasp the relevance of the Jacobian regularization and do not see why the drop in the magnitude of Jacobian norm would be of interest.

---

> ### Author Response · Authors · 2021-11-17
> **Thank you for your insightful and valuable comments.**
>
> Thank you for your insightful and valuable comments. We have tried our best to improve our paper by taking your comments into consideration. We have revised our paper and please kindly check it. We highlighted the changes (in red text) in the revised version.
>
> We summarize what we have updated in the 1st revised version as follows:
> - Notations
>    - To avoid confusion, we use the notation $\langle a\rangle$ for the expected value $\mathbb{E}_\mathcal{D}[a]$.
>    - We use the notation $||A||$ for the operator norm which is equivalent to the spectral norm $||A||_\sigma$ for square matrices.
> - Theorem
>    - The previous discussions on $||J1||$ (or $||\langle J\rangle 1||$) lack theoretical grounding.
>    - Thus, we provide another approach with the operator norm $||\langle J\rangle||^2$ ("the Jacobian norm") with a new Theorem (Thm 2).
>    - We also provide how to estimate the operator norm of the Jacobian in Appendix E (power iteration).
> - EJR
>    - We propose two efficient variants of EJR to train a larger model (WRN-28-10) as detailed in Appendix N.
>    - For a larger model, theses EJR show noticeable performance improvements (Tab 1 and Fig 7).
> - IJR
>    - We analyze the operator norm $||\langle J\rangle||$ and the implicit bias in SGD.
>    - Unlike $||\langle J\rangle 1||$, the operator norm $||\langle J\rangle||$ does not show a step-like structure with plateau, but it still shows three phases of IJR.
>    - We provide a plot (Fig 5) of IJR with loss and accuracy.
> - MSE
>    - In Appendix M, we provide a discussion on the sharpness and the Jacobian norm when trained with the MSE loss instead of the cross-entropy loss.

---

> ### Author Response · Authors · 2021-11-17
> **Answer 1/1**
>
> Before we start, we would like to summarize the paper.
> - Deep neural networks are trained using SGD with discrete step sizes.
> - This discrete step size and the catapult mechanism limit the sharpness below a threshold.
> - As shown in the Hessian decomposition, this implicit effect on the sharpness also affects and regularizes the Jacobian norm.
> - We argue that this effect on the Jacobian norm is a main reason why SGD leads to minima that generalize well.
> - To support this claim, we explicitly regularized the Jacobian norm and observed significant performance improvement.
>
> ------
> We use new reference numbers for Figures, Tables, Sections, etc. used in the 1st revised version. Please kindly check the revised manuscript.
>
> C2…. This is the matrix of an exponential family model that completely ignores all the layers of a neural network apart from the last one where the softmax operates. ...
> - A2
>    - We did not ignore any network parameter. They are considered in the Jacobian $J=\nabla_\theta z$. Note that $p(x;\theta)=\text{softmax}(z(x;\theta))$.
>    - $x\rightarrow z\rightarrow p\rightarrow l$
>    - The first two functions $x\rightarrow z\rightarrow p$ are considered in the Jacobian, and the last one $p\rightarrow l$ in the logit Hessian.
> We decompose the (total) Hessian $H$ into the Jacobian $J$ and the logit Hessian $M$, and investigate the (low-dimensional) logit Hessian to understand the relation between the (total) Hessian $H$ and the Jacobian $J$.
>    - In particular, because of the discrete step size (lr=$\eta$) of SGD and the catapult mechanism, the sharpness is limited near below a threshold. Thus, the Jacobian norm $||\langle J\rangle||$ is regularized accordingly. The regularization effects diminish as the impurity decreases. We argue that this Jacobian regularization is the reason why SGD is biased to minima that generalize well.
>    - To support this main claim, we explicitly regularized the Jacobian norm and observed significant performance improvement.
>
>
> C3: Figure 3 and opening of Section 5 is unclear and too informal for a conference publication. The appendix is optional and findings should be described clearly in the main part of the paper.
> C5: Eq. (11): I did not follow the derivation for this?
> - A(3+5)
>    - We agree that the paper was too informal in a number of places.
> Therefore, in the revised version, we provide a formal derivation of a new sharpness-Jacobian-impurity relation as shown in Eq (11) in Theorem 2.
>    - We have tried our best to address all the raised concerns. Please point out any missing part that requires a further formal discussion.
>
> C4: Section 5.1: Impurity here refers to the top eigenvalue of M? This notion is not defined, there is just a bound of the top eigenvalue in terms of Gini impurity.
> - A4: Yes, it was defined at the end of Sec 4.1, "the top eigenvalue $\lambda^{(1)}$ is bounded by $\frac{1}{2}\text{Gini}(p_{(1)})\leq \lambda^{(1)}\leq \text{Gini}(p_{(1)})$, and thus we call it impurity."
>
> C5: Section 5.2: Could you please clarify the implication that weight norm increase leads to increase in the Jacobian norm? The Jacobian norm should be about the geometry of the loss landscape which can be independent of the actual parameter values? Alternatively, if you start with all-zero parameters then this is to be expected.
> - A5.
>    - It can be intuitively explained, starting with an example of a simple 2-layer network, $z(x)=W^{(2)} D^{(1)}W^{(1)}x$, i.e., $z\_i(x)=W^{(2)}\_{i,:} D^{(1)}W^{(1)}x$.
> We have $\nabla\_{W^{(2)}\_{i,:}} z\_j = \delta\_{i,j}D^{(1)}W^{(1)}x$ which has a norm related to $W^{(1)}$.
> Similarly, the derivative terms with respect to $W^{(1)}$ have a norm related to $W^{(2)}$.
> Inductively, the derivative of the logit with respect to $W^{(l)}$ has a norm related to the other layers’ weight $W^{(i)} (i\neq l)$.
>    - For a deep neural network, a logit element $z_i$ can be written as a sum of products of $a\_1a\_2\cdots a\_n$ where $a_i$’s are some weight elements from each layer (one from each layer) and the input elements. Thus, it is clear that the layerwise weight norm increase is likely to lead to increase in the Jacobian norm $||J||$.
> Like weight decay, we can consider the Jacobian regularization as a model capacity control.
>
> C6: The reasoning on the Jacobian regularization is unclear. Would not Jacobian norm need to decrease for GD to be able to converge?
> - A6
>    - When trained with the cross-entropy, the model parameter $w$ keeps increasing [1] and does not converge in parameter space as shown in Appendix F.
> The convergence in parameter space has nothing to do with the convergence of the training loss or of the test accuracy.
>
> [1] Soudry et al., 2017, The implicit bias of gradient descent on separable data (https://arxiv.org/abs/1710.10345)

---

> > ### Author Response · Authors · 2021-11-23
> > **Answer 2/2**
> >
> > C1. Theorem 1: What is novel in the claim and proof compared to prior work? In particular, I am referring here to the contributions from linear algebra by Golub (1973), Golub & van Loan, and Bunch et al. (1978). For example, Eq. (9) is known as the secular equation of the eigen-decomposition problem for rank-one modification of a diagonal matrix. Results (a-c) are for sure quite well-known in that literature (this might be a misrepresentation of the theoretical contributions). So, the novel contribution here might only be (d), which should then be formulated as a separate proposition and Theorem 1 should be cited with Golub and others.
> > - A1.
> >    - One of our novel contribution is that we investigate the evolution of the top eigenvalue of the logit Hessian during training in an analytical way.
> >    To be specific, the top eigenvalue $\lambda^{(1)}$ is bounded by the Gini Impurity. Thus, we named the top eigenvalue $\lambda^{(1)}$ as __impurity__. It acts like a regularization coefficient for the implicit Jacobian regularization.
> >    - We represent the logit Hessian $\nabla^2_z l$ as a rank-one modified matrix $\text{diag}(p)-pp^T$ and then apply the theory of the eigendecomposition of the rank-one modified matrix.
> >    - The main results are (c) and (d). We used these two results in the later sections.
> >    - The results (a) and (b) are mostly used in Appendix D. We listed them together for brevity.
> >    - We mentioned Golub and others in Theorem 1 (2nd revised version).
> >
> > C5: Section 5.2: Could you please clarify the implication that weight norm increase leads to increase in the Jacobian norm? The Jacobian norm should be about the geometry of the loss landscape which can be independent of the actual parameter values? Alternatively, if you start with all-zero parameters then this is to be expected.
> > - A5'. further comments in addition to A5
> >    -  For a deep neural network, $z=z^{(L)}=W^{(L)}\hat{z}^{(L-1)}$ (the bias term is merged with the weight).
> >    - $\hat{z}^{(0)}=x$
> >    - $z^{(i)} = W^{(i)}\hat{z}^{(i-1)}$ where $i\in[L]=$ {$1,\cdots,L$}  (the bias terms are merged with the weight).
> >    - $\hat z^{(i)} = \sigma(z^{(i)})$ where $\sigma$ is a nonlinear activation function (e.g. ReLU) and $i\in[L-1]$
> >    - $\nabla\_{W^{(l)}\_{i,j}}z_k = \hat{z}^{(l-1)}\_j\nabla\_{{z}^{(l)}\_i} z_k$ where $l\in[L]$ and $k\in[C]$
> >    - Therefore, we have $||J|\|_F^2= \sum\_{i,j,k,l}(\nabla\_{W^{(l)}\_{i,j}}z_k)^2 = \sum\_{i,j,k,l}(\hat{z}^{(l-1)}\_j\nabla\_{{z}^{(l)}\_i} z_k)^2$.
> >    - In other words, the hidden layer activations $\hat{z}^{(l)}$ and the derivatives of the inner layer operations $\nabla\_{{z}^{(l)}} z$ directly affect the Frobenius norm of the Jacobian.
> >    - Similarly, the Jacobian norm $||J||$ is affected by them since $||J||^2\geq ||J|\|^2_F/C$

---

### Official Review · Reviewer_Z9ex · 2021-11-05

**Correctness:** 3
**Technical Novelty And Significance:** 2
**Empirical Novelty And Significance:** 2
**Recommendation:** 3
**Confidence:** 4

**Main Review:**

**Strengths:**
- While not totally novel, it is still a nice, relatively unexplored, approach to look at the low-dimensional logit Hessian to derive some insights.
- In particular, Theorem 1 provides an interesting relation between eigenvalues of (per-sample) logit Hessian and corresponding probability output. This also provides a way to get some bounds on the top eigenvalue of the logit-Hessian (impurity) in terms of the largest probability for a class. The corresponding figures are also quite nice and convey the point.
- The sharpness-impurity-jacobian behaviour and its connection to oscillatory, catapult aspects and others are interesting. (But unfortunately, unclear how confidently we can take away the message, see below. Also, the fact that such a relation between sharpness-impurity-jacobian exists is not too far-fetched, given the form of the Gauss-Newton term.)


**Weaknesses:**
1. Theoretically unmotivated usage of the particular 'Jacobian norm': In Section 5, the authors propose an approximate relation between the Hessian spectral norm, 'impurity', and the 'Jacobian norm'. However, the Jacobian norm is not the usual Jacobian Frobenius norm -- as one would expect or as used in related work (Hoffmann et. al. 2019) -- but is the 'norm' with the particular vector all-one vector (normalized to unit length). This usage is far from being innocuous. It seems that, ideally, the authors would have liked to have the eigenvectors of the logit Hessian, and in particular, q^(1), for the spectral-norm, as also evident from the discussion in the appendix --- but this is perhaps hard to come by practically. Hence the issue is that, in turn, the particular normalized all-one vector used is nothing but the eigenvector corresponding to the zero eigenvalue of the logit Hessian. This is very strange given the particular relationship they want to derive.
Thus, it seems to be an unmotivated heuristic which "surprisingly" seems to hold in practice as the figure 3 shows. (On that note, in Figure 3, the ||H||_\sigma is the spectral norm of the entire Hessian or the Gauss-Newton term G alone? Over how many samples is it computed?)

2. Explicit regularization: Here, the authors do not use their version of the 'Jacobian norm', but rather just the Frobenius norm. Yes, both are closely related and perhaps ultimately regularizing either might have similar effects. But, why is such a choice made in the first place, while not following their own derivation? What happens if you explicitly regularize by your particular Jacobian norm ||J 1||^2? Otherwise, as such, it currently makes the whole 'implicit Jacobian regularization' slightly fishy.

3. Behaviour of the actual Jacobian Frobenius norm: In Figure 3 and 5, it would be very relevant to see how does the Jacobian (Frobenius) norm behaves, not just the particular version of the 'Jacobian norm' used by the authors. Can you please add these to the figures?

4. What does the active regularization period correspond to in terms of the progress in loss or performance? This implicit Jacobian regularization claimed by the paper is only active with a particular (early) phase of training but does not indicate the values of the loss there (please provide them) --- thus, it is hard to concretely understand how 'early' is this phase as well as the fact that why or if it suffices to be present only in this phase?

5. Applicability is restricted to only cross-entropy (CE) loss: An important downside of their approach is that this only applies for CE loss, not something like MSE --- where, the logit-Hessian-equivalent is just the identity matrix. Thus, the offered explanations -- if correct -- will not describe the overall behaviour of generalization in neural networks, only the specific case of CE. I think the authors should very explicitly mention this downside.

6. Other puzzling empirical aspects:
- Regarding the connection between the Jacobian and Fisher penalty, at what point are get Figures in appendix 11 computed? When is this valid in general? Is this valid, at least in the entire 'active regularization period'?
- Figure 7: Here most of the performance gains, with explicit regularization, are 5-10% which, in my opinion, are pretty large to be true. So, i would like to understand this better: what are the particular networks considered (especially the ones on the bottom-left side)? are the networks trained long enough to a similar level of loss (not just fixed # of epochs) so as to render them comparable? I believe the "real" boost, if so, is represented by the point in the upper-right of Figure 7 (right) --- but I would love if my belief is proved wrong.
- Figure 6: If i read the plot correctly, the trend of impurity, as well as the 'Jacobian-norm', for increasing LR and decreasing batch sizes seems to be reversed --- which is not clear why?

7. Presentation of the Sharpness-impurity-jacobian relation: This relation is also vaguely/poorly expressed.
- In Fig 3, when you say \lambda^(1) and J are averaged over the training set, what do you precisely mean in the context of the 'Jacobian-norm'? Is it that the Jacobian J is averaged over the training set and then || E(J) 1||^2 is computed or is it the expectation of the norm over the training set, i.e., E(||J_x 1||^2)? Then why is one preferred over the other? Do they have similar behaviour?
- Given that you refer to \lambda^(1) as per-sample impurity in the previous section, and here implicitly the average over training set is considered. Thus, I ask the authors to properly delineate these two usages, say via the horizontal bar at the top.

8. Unclear/incorrect statements:
- The authors say that Gauss-Newton approximation sometimes fails in the later phase of training. However, I think they are being imprecise here, and strictly speaking, Gauss-Newton approximation (for the Hessian as a whole) can be justified only towards this late phase or end of training. See Sagun et. al., 2017 page 5 "At a point close to a local minimum ....". I guess what the authors mean to say this, only in the context of the top-eigenvalue, that in the late-phase the top-eigenvalue is not clearly isolated from the bulk (which makes sense). So, the authors should really clarify that top-eigenvalue approximation from the Gauss-Newton is valid only during early phase, rather than (incorrectly) conflate explanations as presently done.
- " However, the generalization gap keeps increasing, which provides another counter-example that flat minima generalizes poorly": Where are the corresponding train and test loss curves based on which you make this inference?
- Figure 3 caption is also a bit weird. Can you specify what is the default optimizer, dataset, network? otherwise only some mixed and partial information is present.


**Summary Of The Paper:**

The paper investigates properties of the logit Hessian spectrum, how it relates to the top-eigenvalue of the Hessian (which they refer to as 'sharpness' though it can be debated), and implies a particular implicit regularization of the Jacobian during an early phase of training (which is shown to yield better generalization). They use the derived relations to 'explain' previous observations on sharpness and generalization.

**Summary Of The Review:**

The analysis of the logit Hessian spectrum, in itself, is very interesting. But other than that, most of the other contributions are not so far-fetched or novel, the claim about implicit regularization is not motivated theoretically, many of the empirical aspects are unexplained and questionable, the approach is very specific to cross-entropy loss and thus likely not a general explanation if so. As a result, overall it is unclear if their offered explanations are conclusive and not speculative.

---

> ### Author Response · Authors · 2021-11-16
> **Thank you for your insightful and valuable comments.**
>
> Thank you for your insightful and valuable comments. We have tried our best to improve our paper by taking your comments into consideration.
> We have revised our paper and please kindly check it. We highlighted the changes (in red text) in the revised version.
>
> We summarize what we have updated in the 1st revised version as follows:
> - Notations
>    - To avoid confusion, we use the notation $\langle a\rangle$ for the expected value $\mathbb{E}_\mathcal{D}[a]$.
>    - We use the notation $||A||$ for the operator norm which is equivalent to the spectral norm $||A||_\sigma$ for square matrices.
> - Theorem
>    - The previous discussions on $||J1||$ (or $||\langle J\rangle 1||$) lack theoretical grounding.
>    - Thus, we provide another approach with the operator norm $||\langle J\rangle||^2$ ("the Jacobian norm") with a new Theorem (Thm 2).
>    - We also provide how to estimate the operator norm of the Jacobian in Appendix E (power iteration).
> - EJR
>    - We propose two efficient variants of EJR to train a larger model (WRN-28-10) as detailed in Appendix N.
>    - For a larger model, theses EJR show noticeable performance improvements (Tab 1 and Fig 7).
> - IJR
>    - We analyze the operator norm $||\langle J\rangle||$ and the implicit bias in SGD.
>    - Unlike $||\langle J\rangle 1||$, the operator norm $||\langle J\rangle||$ does not show a step-like structure with plateau, but it still shows three phases of IJR.
>    - We provide a plot (Fig 5) of IJR with loss and accuracy.
> - MSE
>    - In Appendix M, we provide a discussion on the sharpness and the Jacobian norm when trained with the MSE loss instead of the cross-entropy loss.
>
> We will provide further detailed answer for each comment within a few days.

---

> ### Author Response · Authors · 2021-11-17
> **Answer 1/2**
>
> We use new reference numbers for Figures, Tables, Sections, etc. used in the 1st revised version. Please kindly check the revised manuscript.
>
> C1. Theoretically unmotivated usage of the particular 'Jacobian norm'
> - A1.
>    - The Jacobian norm was meant to represent the operator norm of the Jacobian, $||\langle J\rangle||$, and it was approximated by $||\langle J \rangle 1||$ for a computational issue.
> We agree that this is not a good proxy.
> Therefore, in the revised version, we estimate **the operator norm $||\langle J\rangle||$** (we call this the Jacobian norm) by using the power iteration (see Algorithm 2 in Appendix E.7) and derive a **new sharpness-Jacobian-impurity relation (Theorem 2)** which motivates the usage of the operator norm.
>    - In Explicit Jacobian Regularization (EJR), we used the Frobenius norm $||\langle J\rangle||_F$ instead of the operator norm, because it is an **upper bound** of the operator norm $||\langle J\rangle||$, i.e. $||\langle J\rangle||\leq||\langle J\rangle||_F$ and it can **relieves the computational burden** to compute the operator norm $||\langle J\rangle||$ and back-propagate through it (see Algorithm 2 in Appendix E.7).
>    - $||H||_\sigma$ is computed with 5-25\% of the total training set (e.g. 2500 for CIFAR-10) using PyHessian (see Appendix E.5).
> We use the entire Hessian, not the Gauss-Newton term $G$.
>
> C1. Explicit regularization
> - A2
>    - We explain why we used the Frobenius norm for EJR in A1.
>    - We have tried $||\langle J \rangle 1||$ which also showed improvements, but not as good as the Frobenius norm.
>    - We report the performance of Baseline / $||\langle J \rangle 1||$- / $||\langle J \rangle||_F$-regularizations.
>        - CIFAR-10, lr=0.03: 69.83 / 71.04 / 75.53
>        - CIFAR-10, lr=0.1: 70.33 / 71.3 / 74.29
>
> C3. Behaviour of the actual Jacobian Frobenius norm
> - A3. We instead show behaviour of the Jacobian (**operator**) norm in Fig 3,5,6.
>
> C4. What does the active regularization period correspond to in terms of the progress in loss or performance? This implicit Jacobian regularization claimed by the paper is only active with a particular (early) phase of training but does not indicate the values of the loss there (please provide them) --- thus, it is hard to concretely understand how 'early' is this phase as well as the fact that why or if it suffices to be present only in this phase?
> - A4. We plot the test accuracy and the train/test loss together in Figure 5.
> It shows that the gap between the test loss and the train loss is increasing, as the Jacobian norm increases.
> For example, when using SGD with batch size of 128 and lr = 0.01, the sharpness reaches the threshold after about 2 epochs (active regularization period begins) and the regularization effects gradually diminish after about 10-20 epochs (4000-8000 steps).
>
> C5. Applicability is restricted to only cross-entropy (CE) loss
> - A5. We also provide a discussion on MSE loss in Appendix M.
>
> C6-1. Regarding the connection between the Jacobian and Fisher penalty
> - A6-1. For the histogram, we used a model trained for 2000 steps. We also test other models (trained for 500, 1000, 4000 steps) and obtain similar results.
>
> C6-2. Figure7: Here most of the performance gains, with explicit regularization, are 5-10% ...
> - A6-2.
>    - We changed "the previous Figure 7" to Table 1 which shows more detailed results.
>    - For SimpleCNN, we trained both models with and without EJR until we got zero training loss and saturated test accuracy.
> For example, we did full-batch training for 10k-100k epochs and 128-batch training for 100-1000 epochs (40k-400k steps).
> As we used a small network and did not introduce the data augmentation in this case, it is long enough to reach a convergence.
>    - For WRN-28-10, we trained them with the same fixed epoch (e.g. 200, 400, 600).
>
> C6-3. Figure 6: If i read the plot correctly, the trend of impurity, as well as the 'Jacobian-norm', for increasing LR and decreasing batch sizes seems to be reversed --- which is not clear why?
> - A6-3.
>    - If we misunderstood what you meant by "reversed", please kindly correct us.
>    - For the trend of impurity, it shows n-shaped evolution.
> It only represent the progress of training, i.e., a large LR and a large batch lead to a fast training and require a small number of **steps** (not epochs) to reach the maximum (~0.3) and to decrease to a certain value (e.g. 0.2). In other words, the width of plots along x-axis is not the focus in Figure 6.
> We plot impurity together to see if the regularization effects diminish as $\langle\lambda^{(1)}\rangle$ decreases.
>    - For the Jacobian norm, when trained with a large lr and a smaller batch size (dotted), it has a lower value in Active Regularization Period.

---

> > ### Author Response · Authors · 2021-11-17
> > **Answer 2/2**
> >
> > C7-1. Is it that the Jacobian J is averaged over the training set and then || E(J) 1||^2 is computed or is it the expectation of the norm over the training set, i.e., E(||J_x 1||^2)? Then why is one preferred over the other? Do they have similar behaviour?
> > - A7-1. We used $|| \langle J\rangle ||^2$ because of the computational issue. $ \langle|| J ||\rangle^2$ requires computation ($\times|\mathcal{D}|$) of the operator norm for each $J$.
> >
> > C7-2. horizontal bar
> > - A7-2. Thank you for the suggestion. We use the notation $\langle a \rangle$ for the expectation of $a$ over the training set $\mathcal{D}$.
> > This notation is also applicable when $a$ is not a single variable (e.g. $\langle JMJ^T \rangle$).
> >
> > C8-1. Gauss-Newton approximation sometimes fails in the later phase
> > - A8-1. Thank you for the suggestion. We update the expression (top-eigenvalue approximation) accordingly.
> >
> > C8-2. " However, the generalization gap keeps increasing, which provides another counter-example that flat minima generalizes poorly": Where are the corresponding train and test loss curves based on which you make this inference?
> > - A8-2. See A4

---

### Author Response · Authors · 2021-11-16
**revision (1)**

We summarize what we have updated in the 1st revised version as follows:
- Notations
   - To avoid confusion, we use the notation $\langle a\rangle$ for the expected value $\mathbb{E}_\mathcal{D}[a]$.
   - We use the notation $||A||$ for the operator norm which is equivalent to the spectral norm $||A||_\sigma$ for square matrices.
- Theorem
   - The previous discussions on $||J1||$ (or $||\langle J\rangle 1||$) lack theoretical grounding.
   - Thus, we provide another approach with the operator norm $||\langle J\rangle||^2$ ("the Jacobian norm") with a new Theorem (Thm 2).
   - We also provide how to estimate the operator norm of the Jacobian in Appendix E (power iteration).
- EJR
   - We propose two efficient variants of EJR to train a larger model (WRN-28-10) as detailed in Appendix N.
   - For a larger model, theses EJR show noticeable performance improvements (Tab 1 and Fig 7).
- IJR
   - We analyze the operator norm $||\langle J\rangle||$ and the implicit bias in SGD.
   - Unlike $||\langle J\rangle 1||$, the operator norm $||\langle J\rangle||$ does not show a step-like structure with plateau, but it still shows three phases of IJR.
   - We provide a plot (Fig 5) of IJR with loss and accuracy.
- MSE
   - In Appendix M, we provide a discussion on the sharpness and the Jacobian norm when trained with the MSE loss instead of the cross-entropy loss.

---

> ### Author Response · Authors · 2021-11-23
> **revision (2)**
>
> - We mention Golub and others in Theorem 1.

---

### Public Comment · ~Sungyoon_Lee1 · 2023-06-29
**link to published version**

We thank the reviewers again. With their insightful and valuable comments, the contents and the clarity of our paper are much improved in the revised version. Please check our published version at the following link:
https://openreview.net/forum?id=VPiB4Eq4Sx

---

### Decision · Program_Chairs · 2022-01-20

**Decision:**

Reject

**Comment:**

The paper studies an interesting question of the relationship between the eigenvalues of the Hessian matrix with the probability of the output in the logistic loss, and use this to propose a regularization that can improve the performance of the neural networks.

 All the reviewers agree that although the question is interesting, the paper lacks significantly in terms of representation and would benefit from another round of revision.